https://doi.org/10.1038/s41467-021-21853-6　　**OPEN**

# Versatility in acyltransferase activity completes chicoric acid biosynthesis in purple coneflower

Rao Fu [1,5], Pingyu Zhang[1,5], Ge Jin[1], Lianglei Wang[1], Shiqian Qi[2], Yang Cao[3], Cathie Martin [4] & Yang Zhang [1✉]

Purple coneflower (*Echinacea purpurea* (L.) Moench) is a popular native North American herbal plant. Its major bioactive compound, chicoric acid, is reported to have various potential physiological functions, but little is known about its biosynthesis. Here, taking an activity-guided approach, we identify two cytosolic BAHD acyltransferases that form two intermediates, caftaric acid and chlorogenic acid. Surprisingly, a unique serine carboxypeptidase-like acyltransferase uses chlorogenic acid as its acyl donor and caftaric acid as its acyl acceptor to produce chicoric acid in vacuoles, which has evolved its acyl donor specificity from the better-known 1-*O*-β-D-glucose esters typical for this specific type of acyltransferase to chlorogenic acid. This unusual pathway seems unique to *Echinacea* species suggesting convergent evolution of chicoric acid biosynthesis. Using these identified acyltransferases, we have reconstituted chicoric acid biosynthesis in tobacco. Our results emphasize the flexibility of acyltransferases and their roles in the evolution of specialized metabolism in plants.

[1] Key Laboratory of Bio-resource and Eco-environment of Ministry of Education, College of Life Sciences, Sichuan University, Chengdu 610064, China. [2] Department of Urology, State Key Laboratory of Biotherapy, West China Hospital, Sichuan University, and Collaborative Innovation Center for Biotherapy, Chengdu 610041, China. [3] Center of Growth, Metabolism and Aging, College of Life Sciences, Sichuan University, Chengdu 610064, China. [4] Department of Metabolic Biology and Biological Chemistry, John Innes Centre, Norwich NR4 7UH, UK. [5]These authors contributed equally: Rao Fu, Pingyu Zhang. ✉email: yang.zhang@scu.edu.cn

The discovery of America by Columbus changed the world in many ways, particularly through the increased exchange of valuable natural resources. One example was the introduction of *Echinacea*, a genus of herbs native to North America, and used by Native Americans as traditional medicines to treat common colds and infections[1]. Since the 1880s, *Echinacea* extracts have become some of the most popular commercial herbal preparations, and today over 1000 products containing extracts of *Echinacea* are listed in the Dietary Supplement Label Database (https://dsld.od.nih.gov). The total sales of products from *E. purpurea*, *E. angustifolia,* and *E. pallida*, reached US\$ 120.2 million on the US market alone in 2019 and grew by 90.9% in the first half of 2020, partially due to the COVID-19 pandemic[2]. The main bioactives in *Echinacea* are caffeic acid derivatives (CADs)[3]. Chicoric acid is the principal CAD and provides an index for the quality of raw material and commercial preparations of *Echinacea*[4]. Although there isn't sufficient evidence to conclusively establish its functions, chicoric acid has been reported as an potential HIV-1 integrase inhibitor[5] along with other bioactivities including antiviral, anti-inflammatory, glucose and lipid homeostatic, neuroprotective, anti-atherosclerosis, and anti-aging activities[6]. Despite the large number of studies in the potential health benefits of chicoric acid, its biosynthetic pathway remains unclear.

Enzymes that use caffeoyl CoA and *meso*-tartaric acid to generate mono-*O*-caffeoyl-*meso*-tartrate and add an additional caffeoyl group from caffeoyl CoA to form di-*O*-caffeoyl-*meso*-tartrate were reported in *Equisetum arvense* in 1996, but without identification of the encoding genes[7]. In *Echinacea*, caftaric acid and chicoric acid are L-isomers, which differ from those in *Equisetum arvense* suggesting a distinct biosynthetic pathway[8]. A hydroxycinnamoyl-CoA: tartaric acid hydroxycinnamoyl transferase (HTT) activity synthesizing L-caftaric acid has been reported in *Arachis glabrata*, but without gene information[9]. However, no enzyme activity has yet been described for L-chicoric acid biosynthesis. One of the best-studied CADs is chlorogenic acid. Hydroxycinnamoyl-Coenzyme A: quinate hydroxycinnamoyl transferase (HQT), a BAHD acyltransferase family member, is involved in the biosynthesis of chlorogenic acid in different plant species[10–12]. Another important acyltransferase group is the serine carboxypeptidase-like (SCPL) family. The major differences between these two families are the acyl donor source and cellular compartment: BAHDs use acyl-CoA thioesters as acyl donors in the cytosol, whereas SCPLs use 1-*O*-β-glucose esters as their acyl donors in the vacuole[13].

Here we report the elucidation of the chicoric acid biosynthesis pathway in purple coneflower. This pathway involves both BAHD and SCPL acyltransferases. Two BAHD enzymes, EpHTT and EpHQT are responsible for the biosynthesis of caftaric acid and chlorogenic acid in the cytosol, respectively. These two CADs serve as substrates for the generation of chicoric acid in the vacuole by a unique SCPL enzyme. Unlike other SCPLs, this SCPL prefers chlorogenic acid to 1-*O*-caffeoyl β-D-glucose as its acyl donor. Among the species we checked, this chicoric acid biosynthetic route is unique to *Echinacea spp.* and it is likely that chicoric acid biosynthesis in other species evolved by convergent catalytic mechanisms. Our results expand the understanding of acyltransferases and highlight the rapid evolution possible with these types of enzymes.

## Results

### EpBAHDs catalyze the synthesis of caftaric acid and chlorogenic acid.
Previous studies indicated that purple coneflower (*Echinacea purpurea*) contains much higher levels of chicoric acid than other *Echinacea* species[14]. Metabolic profiling of purple coneflower seedlings revealed the presence of chicoric acid throughout the plant, making it difficult to conduct tissue-specific coexpression analysis to identify biosynthetic genes (Fig. 1)[15–17]. We generated hairy root cultures of purple coneflower and confirmed the presence of chicoric acid (Fig. 1). RNA-seq of hairy roots treated with methyl jasmonate to induce chicoric acid accumulation failed to provide good candidate genes due to the large number of differently expressed genes. As an alternative, we took a traditional biochemical approach and searched for caftaric acid synthase activity as caftaric acid is a likely intermediate (Fig. 1d)[12,18]. Based on the structural similarity of caftaric acid to chlorogenic acid, we investigated whether hydroxycinnamoyl-CoA: tartaric acid hydroxycinnamoyl transferase (HTT) could use caffeoyl CoA as an acyl donor to produce caftaric acid as reported[9]. When tartaric acid and caffeoyl CoA were incubated with a crude protein extract from hairy roots of purple coneflower, we detected the production of caftaric acid, indicating the presence of HTT activity. In addition, the crude protein extract also catalyzed the generation of chlorogenic acid using quinic acid and caffeoyl CoA as substrates, confirming the activity of hydroxycinnamoyl-Coenzyme A: quinate hydroxycinnamoyl transferase (HQT).

We purified HTT activity by ammonium sulfate precipitation and ion-exchange chromatography and found one candidate protein annotated as HQT-like in the fraction with high HTT activity through comparison of peptide mass fingering data to hairy root RNA-seq data (Supplementary Fig. 1a-b). This protein belongs to the BAHD acyltransferase family, which utilize CoA thioesters as acyl donors: BAHD enzymes have two conserved amino acid motifs HXXXD and DFGWG[11,13]. This HQT-like protein was closely related to the HCT (Hydroxycinnamoyl-Coenzyme A: quinate/shikimate hydroxycinnamoyl transferase) clade, with HHLVD and DFGWG motifs (Fig. 2a and Supplementary Fig. 2). In vitro enzyme assays using recombinant protein from *E. coli* confirmed that this enzyme can catalyze the biosynthesis of caftaric acid (Fig. 2b and Supplementary Fig. 3a and 4a-c), and we named it EpHTT. Similarly, in fractions with high HQT activity, we identified another BAHD acyltransferase with classic HTLSD and DFGWG motifs belonging to the HQT clade of BAHD acyltransferases (Fig. 2a, Supplementary Fig. 1a-b and 2), which could catalyze the formation of chlorogenic acid in vitro, and which we named EpHQT (Fig. 2c and Supplementary Fig. 3b and 4d-f). No putative signal peptides were found in the EpHTT or EpHQT sequences. Both EpHTT and EpHQT GFP-tagged proteins were cytosolically localized as reported for other HQTs[19,20] and showed highest activities at pH 7 (Fig. 2h and Supplementary Fig. 5a-b). Although both enzymes showed promiscuity in their acyl donors, caffeoyl CoA was preferred by both (Supplementary Table 1-2). EpHTT differs from HTT activity from perennial peanut which prefers *p*-coumaroyl CoA as its acyl donor[9]. We verified the function of *EpHTT* and *EpHQT* in vivo using both overexpression and gene silencing in hairy roots. When the *EpHTT* gene was overexpressed or silenced, the caftaric acid levels in transgenic hairy roots increased or decreased correspondingly (Fig. 2d, e). Similar results were seen for the *EpHQT* gene, where the contents of chlorogenic acid in the transgenic hairy roots were correlated positively in the different lines to the expression levels of *EpHQT* (Fig. 2f, g). Our in vitro and in vivo results confirmed that EpHTT and EpHQT are responsible for caftaric acid and chlorogenic acid biosynthesis, respectively in purple coneflower.

### Substrate specificities of EpBAHDs.
Phylogenetic analysis indicated that EpHTT might have evolved from the hydroxycinnamoyl-coenzyme A: shikimate/quinate hydroxycinnamoyl transferase

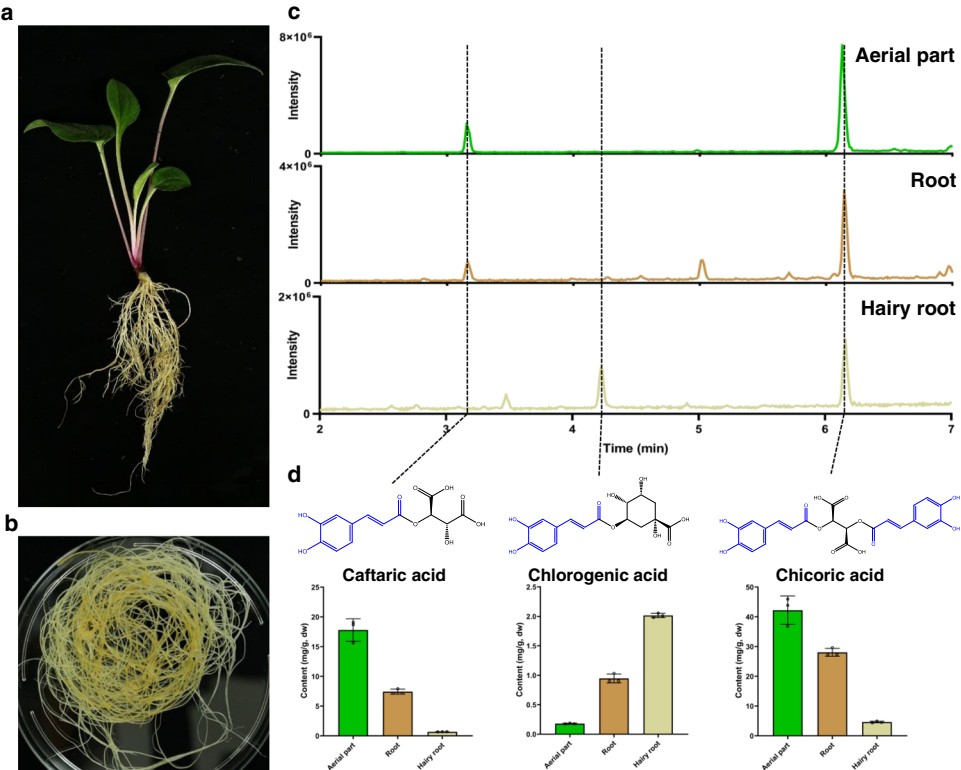

**Fig. 1 Main caffeic acid derivatives in purple coneflower. a** Two-month-old seedlings and **b** hairy root cultures of purple coneflower. **c** Total ion chromatograms (TIC) of aerial parts, roots, and hairy root cultures. **d** Main caffeic acid derivatives (CADs) and their abundance. Data are mean±s.d. ($n = 3$ biologically independent samples). Source data underlying Fig. 1d are provided as a Source Data file.

(HCT) clade of BAHD acyltransferases (Fig. 2a). Using AtHCT (AED95744.1) and NtHCT (CAE46932.1) to screen RNA-seq data, we also identified a candidate transcript encoding EpHCT. EpHCT belongs to the HCT clade with conserved HHAAD and DFGWG motifs (Fig. 2a and Supplementary Fig. 2)[21]. We confirmed EpHCT function by in vitro enzyme assays which could catalyze the production of p-coumaroyl shikimic acid using p-coumaroyl CoA and shikimic acid as substrates, and the generation of 5-O-caffeoyl shikimic acid using caffeoyl CoA and shikimic acid as substrates (Supplementary Fig. 6). All three BAHD enzymes, EpHTT, EpHQT and EpHCT, showed preferences amongst their potential acyl acceptors (Table 1). EpHTT could use only tartaric acid but not quinic acid nor shikimic acid, EpHQT and EpHCT could use both quinic acid and shikimic acid as acyl acceptors but with significant preferences by EpHQT for quinic acid and by EpHCT for shikimic acid as previously reported (Supplementary Fig. 7)[12,21,22]. Molecular docking was undertaken based on the structural model for p-coumaroyl shikimic acid bound AtHCT (PDB: 5KJU)[23]. All three EpBAHDs have similar scaffolds. Compared with EpHQT and EpHCT, EpHTT formed a smaller pocket, that could accommodate only tartaric acid, and not quinic acid and shikimic acid (Supplementary Fig. 8a). Differences between the active pockets of EpHQT and EpHCT appeared critical in determining their acyl acceptor preferences (Supplementary Fig. 8b). In summary, EpHTT has likely evolved from the HCT clade of enzymes, principally through specialization of its acyl acceptor recognition.

**Chicoric acid is synthesized from caftaric acid and chlorogenic acid**. As suggested by the formation of di-caffeoyl quinic acid in tomato and di-O-caffeoyl-meso-tartrate in Equisetum arvense[7,24], we tried to establish a similar in vitro chicoric acid biosynthesis assay. However, neither EpHTT nor crude protein extract could convert caftaric acid or caftaric acid with caffeoyl CoA to chicoric

acid (Supplementary Fig. 9a, b). We noticed that decreases in EpHQT expression also led to the decline of chicoric acid content in transgenic hairy roots (Supplementary Fig. 9c). Chlorogenic acid has been reported as an acyl donor for the biosynthesis of di-caffeoyl quinic acid by SlHQT, as well as caffeoylglucarate by a GDSL lipase-like acyltransferase in tomato[24,25]. This suggested that chlorogenic acid might be used as acyl donor in chicoric acid biosynthesis. Consequently, we incubated caftaric acid and chlorogenic acid with a crude protein extract from purple coneflower and detected the formation of chicoric acid. We optimized this assay using acidic and low salt conditions (PBS pH 4.0 without NaCl or KCl) (Supplementary Fig. 9b). Together, these data suggested that chicoric acid can be synthesized from caftaric acid and chlorogenic acid by a chicoric acid synthase in purple coneflower.

**A SCPL enzyme catalyzes the biosynthesis of chicoric acid**. The activity of chicoric acid synthase was measured and fractions with the highest activities were used for peptide mass fingering (Supplementary Fig. 1c, d). However, peptides of neither BAHDs nor GDSL lipase-like proteins were detected. Instead, we noticed peptides of a Serine carboxypeptidase-like (SCPL) protein were enriched in the fractions (Supplementary Fig. 1e). This protein belongs to the SCPL clade IA which has been reported to have acyltransferase activity (Fig. 3a)[26]. In vitro assays indicated that this protein, when expressed and purified from yeast, could catalyze the trans-acylation of caftaric acid by chlorogenic acid to form chicoric acid and quinic acid (Fig. 3b and Supplementary Fig. 10). Therefore, we named this SCPL protein, chicoric acid synthase (EpCAS). Compared to BAHD acyltransferases, EpCAS required longer incubation times for accumulation of its chicoric acid product to become apparent, which is typical of other SCPLs[27,28]. We determined the kinetic parameters of EpCAS

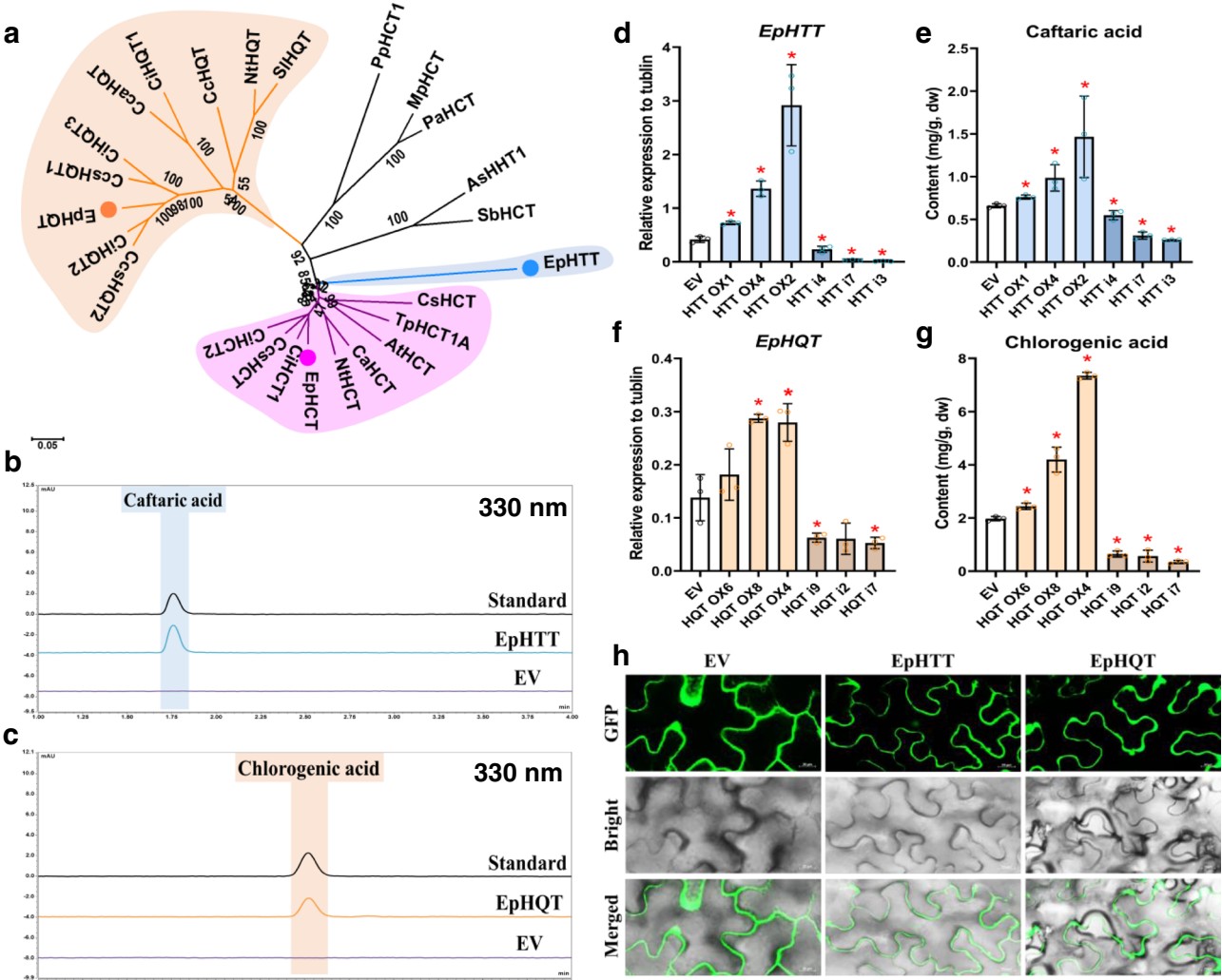

**Fig. 2 Two cytosolic BAHD acyltransferases catalyze the synthesis of caftaric acid and chlorogenic acid. a** Phylogenic analysis of EpHTT, EpHQT, and EpHCT with functionally identified BAHDs. GeneBank ID of the proteins used in the tree are listed in methods. The HQT and HCT clades are highlighted in orange and pink, respectively, and EpHTT is highlighted in blue. UPLC detection of caftaric acid and chlorogenic acid production by EpHTT (**b**) and EpHQT (**c**), respectively. **d** Relative expression levels of *EpHTT* and **e** caftaric acid contents in transgenic overexpression (OX) and RNAi (i) hairy root lines. **f** Relative expression levels of *EpHQT* and **g** chlorogenic acid contents in transgenic hairy root lines. Data are mean ± s.d. (n = 3 biologically independent samples). * indicates significant difference from empty vector (EV) control line (P < 0.05) analyzed by two-sided Student's *t*-test. *P = 0.0013 (HTT OX1), *P = 0.0005 (HTT OX4), *P = 0.0047 (HTT OX2), *P = 0.0163 (HTT i4), *P = 0.0003 (HTT i7), *P = 0.0002 (HTT i3) in **d**; *P = 0.0072 (HTT OX1), *P = 0.0230 (HTT OX4), *P = 0.0430 (HTT OX2), *P = 0.0302 (HTT i4), *P = 0.0002 (HTT i7), *P = 0.000008 (HTT i3) in **e**; *P = 0.0042 (HQT OX8), *P = 0.0118 (HQT OX4), *P = 0.0420 (HQT i9), *P = 0.0298 (HQT i7) in **f**; *P = 0.0053 (HQT OX6), *P = 0.0013 (HQT OX8), *P = 0.0000004 (HQT OX4), *P = 0.0001 (HQT i9), *P = 0.0005 (HQT i2), *P = 0.000009 (HQT i7) in **g**. **h** Cytosolic localization of EpHTT and EpHQT in *Nicotiana benthamiana* leaves. Scale bars show 20 μm. This experiment was repeated independently three times with similar results. Source data underlying d–g are provided as a Source Data file.

using in vitro enzyme assays and found that EpCAS showed quite low *Kcat* values for its caftaric acid and chlorogenic acid substrates (Supplementary Table 3). Similar to other SCPLs[29,30], EpCAS has a conserved vacuolar targeting peptide at its N-terminus (Met1-His24) (Supplementary Fig. 11). We confirmed that an EpCAS GFP-fusion protein was located in vacuoles, and the enzyme shows highest activity at pH 4 consistent with its vacuolar location (Fig. 3e and Supplementary Fig. 5c). We verified the role of EpCAS in chicoric acid biosynthesis in vivo using hairy roots. When *EpCAS* was overexpressed or silenced, the chicoric acid content was increased or decreased respectively, consistent with *EpCAS* expression levels which were accordingly upregulated or downregulated (Fig. 3c, d). These results showed that EpCAS catalyzes the synthesis of chicoric acid in purple coneflower.

When the EpCAS was aligned with other functionally characterized SCPL acyltransferases, we found it had the conserved Ser-His-Asp catalytic triad (Supplementary Fig. 12), which is necessary for hydrolysis of peptide bonds by serine proteinase and can be inhibited by phenylmethylsulfonyl fluoride (PMSF)[31,32]. Indeed, the in vitro activity of EpCAS was inhibited by PMSF in a dose-dependent manner (Fig. 4a). The significance of the conserved Ser-His-Asp catalytic triad of EpCAS was also confirmed using site-directed mutagenesis. When the Ser180 was replaced by Ala, EpCAS lost acyltransferase activity completely. Mutations converting His430 and Asp374 to alanine also showed significantly lower activities (Fig. 4b). These results demonstrated that the catalytic triad is crucial for the acyl transfer activity[33]. Molecular docking based on the structure of the yeast serine

**Table 1 EpBAHD substrate preferences.**

| Enzyme | Acyl donors | Acyl acceptors | | |
| --- | --- | --- | --- | --- |
| | | Tartaric acid | Quinic acid | Shikimic acid |
| EpHTT | Caffeoyl CoA | + | n.d. | n.d. |
| | *p*-Coumaroyl CoA | + | n.d. | n.d. |
| | Feruloyl CoA | + | n.d. | n.d. |
| EpHQT | Caffeoyl CoA | n.d. | + | + |
| | *p*-Coumaroyl CoA | n.d. | + | + |
| | Feruloyl CoA | n.d. | + | n.d. |
| EpHCT | Caffeoyl CoA | n.d. | + | + |
| | *p*-Coumaroyl CoA | n.d. | + | + |
| | Feruloyl CoA | n.d. | n.d. | n.d. |

"+" indicates that the reaction can happen. n.d. means no detection of corresponding products under these experimental conditions.

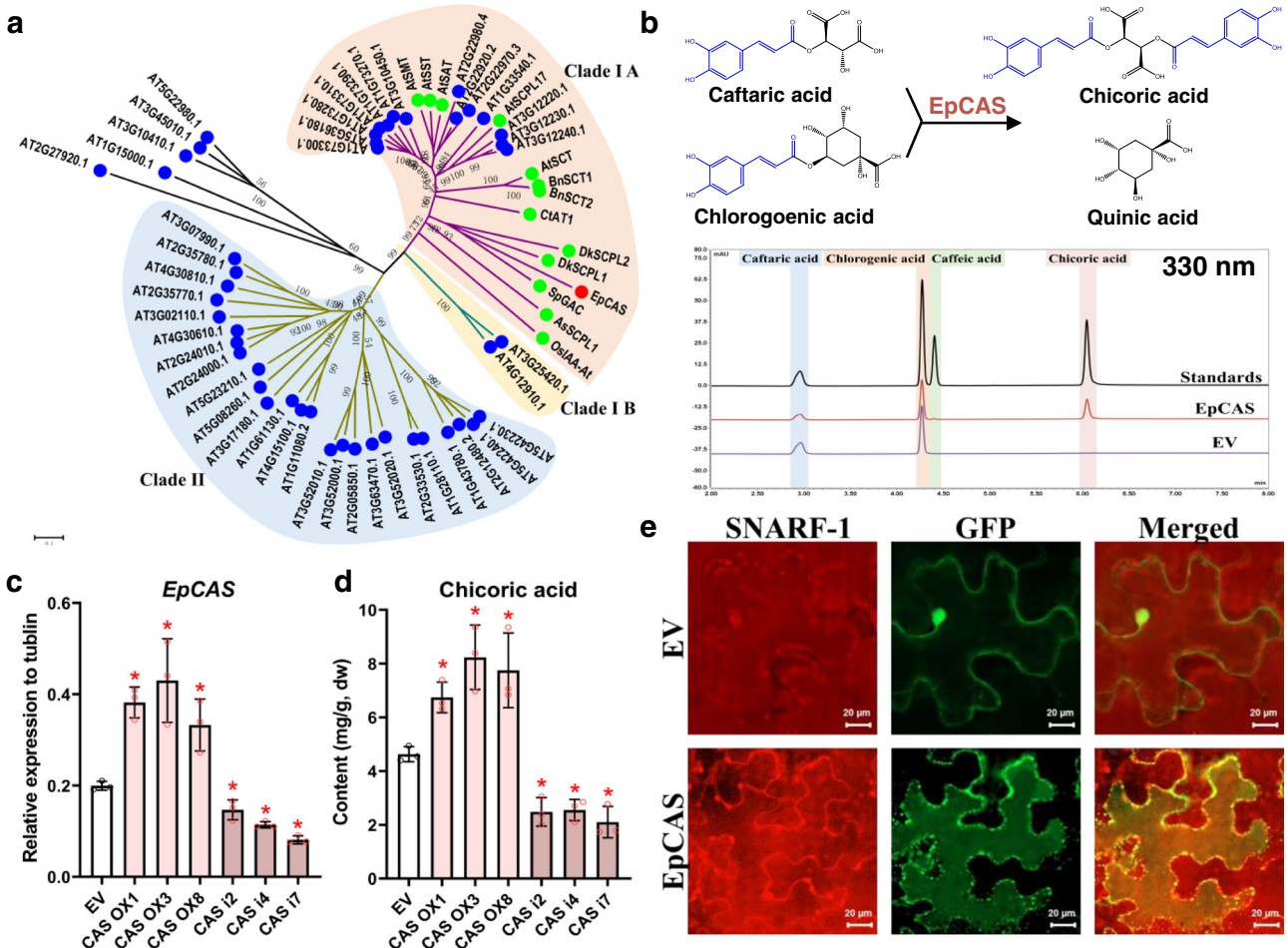

**Fig. 3 A SCPL enzyme catalyzes the biosynthesis of chicoric acid. a** Phylogenic analysis of EpCAS together with *Arabidopsis thaliana* and several additional functionally characterized SCPLs. EpCAS is marked in red. SCPLs previously characterized functionally are marked in green. **b** UPLC detection of chicoric acid generated by recombinant EpCAS. **c** Relative expression levels of *EpCAS* and **d** chicoric acid contents in transgenic over expression (OX) and RNAi (i) hairy root lines. Data are mean±s.d. (*n* = 3 biologically independent samples). * indicates a significant difference from the empty vector (EV) control line (*P* < 0.05) analyzed by two-sided Student's *t*-test. *P* = 0.0008 (CAS OX1), *P* = 0.0123 (CAS OX3), *P* = 0.0162 (CAS OX8), *P* = 0.0187 (CAS i2), *P* = 0.0002 (CAS i4), *P* = 0.0001 (CAS i7) in **c**; *P* = 0.0045 (CAS OX1), *P* = 0.0071 (CAS OX3), *P* = 0.0189 (CAS OX8), *P* = 0.0034 (CAS i2), *P* = 0.0017 (CAS i4), *P* = 0.0025 (CAS i7) in **d**. **e** EpCAS is vacuole-localized in *Nicotiana benthamiana* leaves. The vacuole was visualized by SNARF-1[55]. Scale bars, 20 μm. This experiment was repeated independently three times with similar results. Source data underlying **c** and **d** are provided as a Source Data file.

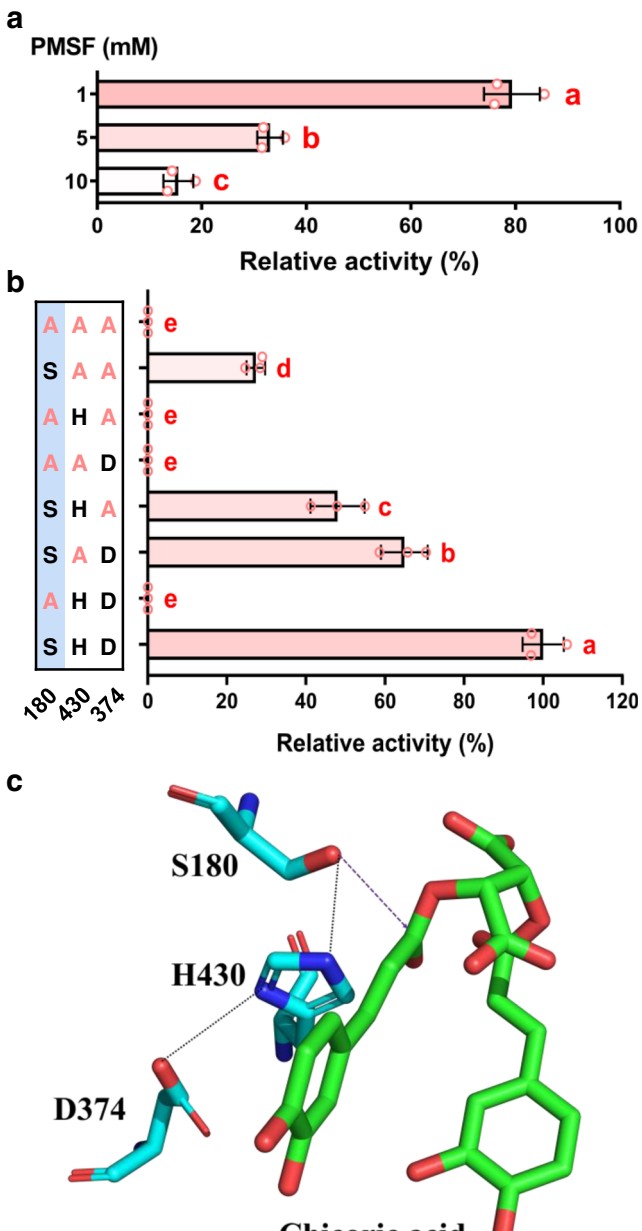

**Fig. 4 The Ser-His-Asp catalytic triad is necessary for the acyltransferase activity of EpCAS. a** EpCAS activity was inhibited by Phenylmethylsulfonyl fluoride (PMSF) in a dose-dependent manner. **b** Site-directed mutagenesis of Ser-His-Asp catalytic triad. Ser180 is necessary for the acyltransferase activity of EpCAS. Data are mean±s.d. ($n = 3$ independent experiments). Different letters in **a** and **b** indicate significantly different values at $P < 0.05$ analyzed by one-way ANOVA with Tukey's multiple comparisons test. **c** Molecular docking of Ser-His-Asp catalytic triad with chicoric acid. Source data underlying **a** and **b** are provided as a Source Data file.

carboxypeptidase (PDB: 1YSC) was used to investigate the catalytic mechanism of EpCAS[34]. The catalytic triad of EpCAS formed the reactive part of the active site, especially Ser180 which could act as a nucleophilic residue and attack the carbonyl C of the chlorogenic acid to form an ester with the caffeic acid intermediate. The ester bond of this complex could be subsequently attacked by caftaric acid and result in the production of chicoric acid (Fig. 4c). Surprisingly, we also detected caffeic acid as a byproduct in the EpCAS reaction (Fig. 3b and Supplementary Fig. 10f-g). An SCPL (1-O-sinapoyl-β-D-glucose: L-malate

sinapoyl transferase) from *Arabidopsis* has also been reported to have a minor hydrolytic activity producing sinapic acid from 1-O-sinapoyl-β-D-glucose (pH 6.0)[35]. When EpCAS was mixed with caftaric acid or chlorogenic acid alone (pH 4.0), it cleaved caffeic acid from both these substrates (Supplementary Fig. 13). Another SCPL from *Zea mays*, 1-O-(indole-3-acetyl)-β-D-glucose: *myo*-inositol indoleacetyl transferase has also been reported to have hydrolytic activity towards its product[36]. EpCAS could also hydrolyze chicoric acid to release caftaric acid and caffeic acid, but this reaction was irreversible (Supplementary Fig. 13c-d). It seems that the Ser180 of the catalytic triad can attack the ester bond of these caffeic acid esters directly and release minor amounts of caffeic acid.

**The substrate specificity of EpCAS.** Since previous studies have indicated that SCPL acyltransferases use 1-O-β-D-glucose esters as their acyl donors[13,37], we investigated whether EpCAS could also use 1-O-caffeoyl-β-D-glucose as its acyl donor. When we incubated 1-O-caffeoyl-β-D-glucose (Supplementary Fig. 14) with caftaric acid, very little chicoric acid was detected (Supplementary Fig. 15a-b). Compared to chlorogenic acid, the relative activity of EpCAS in using 1-O-caffeoyl-β-D-glucose was reduced to $0.67 \pm 0.06\%$ (Supplementary Fig. 16a). The in vitro kinetics of EpCAS against 1-O-caffeoyl-β-D-glucose compared to chlorogenic acid showed that even though they had comparable $Km$ values, the $Kcat$ and $Kcat/Km$ values were much lower for 1-O-caffeoyl-β-D-glucose (Supplementary Table 3). Docking results showed that the glucose group of 1-O-caffeoyl-β-D-glucose would sink into the inner pocket and hinder the entry of caftaric acid. Although quinic acid and glucose have similar structures, the extra carboxyl group of quinic acid would hinder the entrance of chlorogenic acid to the inner pocket of EpCAS. This difference suggested that quinic acid would be liberated more easily from chlorogenic acid than glucose from 1-O-caffeoyl-β-D-glucose, explaining the preference of EpCAS for chlorogenic acid as its acyl donor (Supplementary Fig. 15c). Interestingly, we found that compounds with structures similar to chlorogenic acid (neochlorogenic acid, cryptochlorogenic acid and 5-O-caffeoyl shikimic acid) could also be used by EpCAS but with much lower relative activities (<1%) than with chlorogenic acid under identical conditions (Supplementary Fig. 16). These results demonstrated that EpCAS has evolved unique specificity for its acyl donor, distinct from that of other characterized SCPL enzymes in plants[33,38]. Interestingly, although we could detect very low activity of purified EpCAS enzyme when using 1-O-caffeoyl-β-D-glucose as an acyl donor in vitro, crude protein extracts of purple coneflower could not use 1-O-caffeoyl-β-D-glucose as an acyl donor to synthesize caftaric acid, chlorogenic acid or chicoric acid (Supplementary Fig. 17). In addition, we failed to detect any 1-O-caffeoyl-β-D-glucose in extracts of purple coneflower (Supplementary Fig. 18). All these results demonstrated that in purple coneflower, EpCAS has evolved specificity for its acyl donor.

**Chicoric acid biosynthesis is polyphyletic in the plant kingdom.** So far, over sixty genera and species have been found to contain chicoric acid[39]. To investigate the conservation of EpCAS's unique acyltransferase activity in other chicoric acid producing species, we selected several representative species including: another two *Echinacea* species (*Echinacea pallida* and *Echinacea angustifolia*) and two distantly related species (*Cichorium endivia* and *Lactuca sativa*) known to produce caftaric acid, chlorogenic acid and chicoric acid; as well as *Helianthus annuus* and *Chysanthemum coronarium*, from two different intervening plant lineages which accumulate only chlorogenic acid (Fig. 5a-b and Supplementary Fig. 19). When we incubated

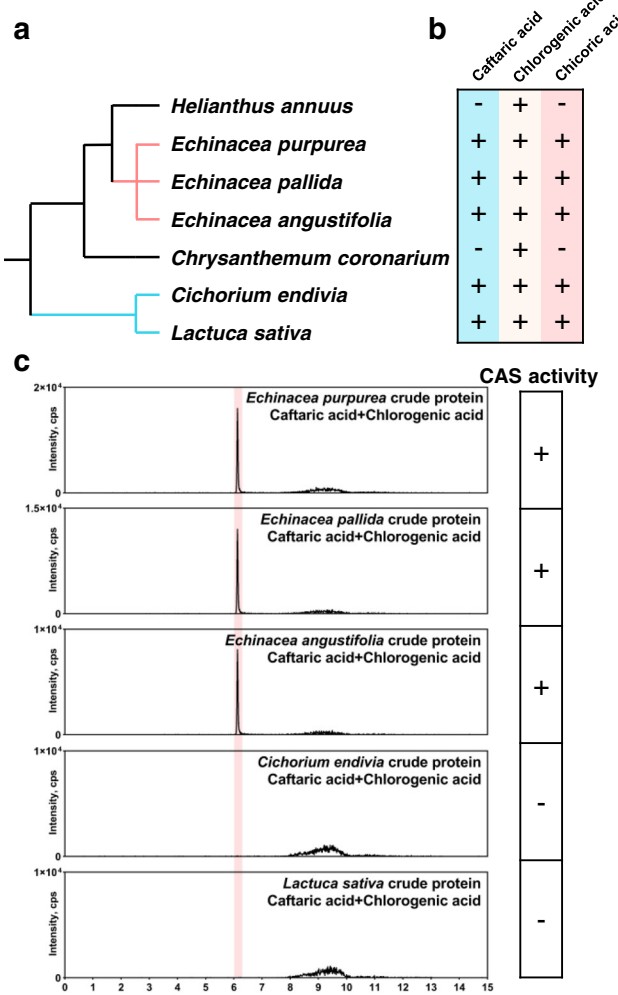

**Fig. 5 Chicoric acid biosynthesis is polyphyletic between different species. a** Phylogenetic relationships among several related species. Two *Echinacea* species (*Echinacea pallida* and *Echinacea angustifolia*) and two distantly related species (*Cichorium endivia* and *Lactuca sativa*) containing both substrates (caftaric acid and chlorogenic acid) and chicoric acid; as well as *Helianthus annuus* and *Chysanthemum coronarium*, from two intervening plant lineages which contain only chlorogenic acid were selected for the study of chicoric acid biosynthetic routes. The tree was adopted from NCBI taxonomy (http://lifemap-ncbi.univ-lyon1.fr/). **b** The presence of target compounds in each species. **c** CAS activities from different chicoric acid containing plant crude protein extracts. Only *Echinacea* species can use caftaric acid as an acyl acceptor and chlorogenic acid as an acyl donor to produce chicoric acid.

crude protein extracts from the other chicoric acid-producing species with caftaric acid and chlorogenic acid, *E. pallida* and *E. angustifolia* crude protein extracts showed CAS activities. However, *Cichorium endivia* and *Lactuca sativa* failed to produce chicoric acid (Fig. 5c). Crude protein extracts from all samples exhibited HQT activities, but *Cichorium endivia* and *Lactuca sativa* crude protein extracts did not show HTT activities agreeing with their lack of CAS activity (Supplementary Fig. 20). In addition, we were unable to find any sequences encoding highly homologous proteins to EpHTT and EpCAS from the *Lactuca sativa* genome or any other public available databases by BLAST, while the BAHD proteins with the HTLSD motif (conserved in HQT-likes proteins) were found. It seems likely that the pathway for chicoric acid biosynthesis catalyzed by HTT and CAS in

*Echinacea* species is unique. In other chicoric acid containing species, we conclude that chicoric acid is likely synthesized using different substrates and enzyme activities. Our results support the idea that chicoric acid biosynthesis is polyphyletic and has evolved in different plant lineages by convergence[40].

**Reconstruction of chicoric acid biosynthesis in *N. benthamiana*.** As a proof of concept, we attempted to reconstruct chicoric acid biosynthesis in planta. As *N. benthamiana* leaves produce abundant chlorogenic acid but no tartaric acid[12], different biosynthetic genes (*EpHTT* and *EpCAS*) and precursor (tartaric acid) combinations were investigated for their ability to produce chicoric acid. Coexpression of *EpHTT* and *EpCAS* and subsequent supplementation with tartaric acid generated chicoric acid verifying the functions of the identified genes and showing that they could be used for engineering chicoric acid production in plant chassis (Fig. 6). Future attempts to scale-up production of chicoric acid could be explored in chassis with a good source of tartaric acid combined with other metabolic engineering tools[41].

## Discussion

For millennia, humans have used plant specialized metabolites as herbal medicines. During the past two decades, especially in the postgenomic era, strategies such as co-expression and genome mining have been used extensively for metabolic pathway identification[42]. In this study, however, we describe steps in a metabolic pathway identified using traditional biochemistry. While RNA-seq, co-expression analysis and genome mining are becoming more popular approaches, discovery of genuinely novel activities may still be served best by strategies based on characterizing enzyme activities[42].

We have shown that the biosynthesis of chicoric acid, the major bioactive compound of purple coneflower, involves a two-step process. In the cytosol, two BAHD acyltransferases, EpHTT and EpHQT catalyze the production of caftaric acid and chlorogenic acid intermediates, respectively. Both compounds are transported to the vacuole to form chicoric acid catalyzed by EpCAS (Fig. 7). In purple coneflower, chicoric acid has been reported to be stored in vacuoles[43], in line with our proposed biosynthetic pathway. Chicoric acid is also present in intercellular spaces and cell walls which may be due to release following cell damage. Like other identified SCPLs[13,35,44], EpCAS prefers acidic conditions and exhibits highest in vitro enzyme activity at pH 4, like SlHQT chlorogenate-chlorogenate transferase activity in vacuoles[24]. EpCAS activity was relatively high (>80% of the maximum) over the pH range of 3.6 to 4.9, which should ensure the production of chicoric acid under vacuolar pH conditions. Based on our results, the biosynthesis of chicoric acid is likely to depend on transport of caftaric acid and chlorogenic acid into the vacuoles, although, to date, no transporters for hydroxycinnamic acid derivatives have been reported. Chlorogenic acid has been reported to be transported into vacuoles by vesicles in *Lonicera japonica* flowers[45].

Chicoric acid has been reported to have multiple physiological functions[6], making it an important target for synthetic biology. Using *N. benthamiana*, we have successfully reconstituted the chicoric acid biosynthetic pathway. However, due to the absence of tartaric acid, *N. benthamiana* is not the ideal chassis for producing chicoric acid. Since the biosynthesis of tartaric acid has not been fully elucidated[46,47], further attempts to reconstruct chicoric acid biosynthesis should be focused on chassis with naturally high tartaric acid contents.

Our results emphasize the flexibility of specialized metabolism in plants, particularly the broad importance of transfer of acyl groups to form specialized bioactives. Unlike other SCPLs, in

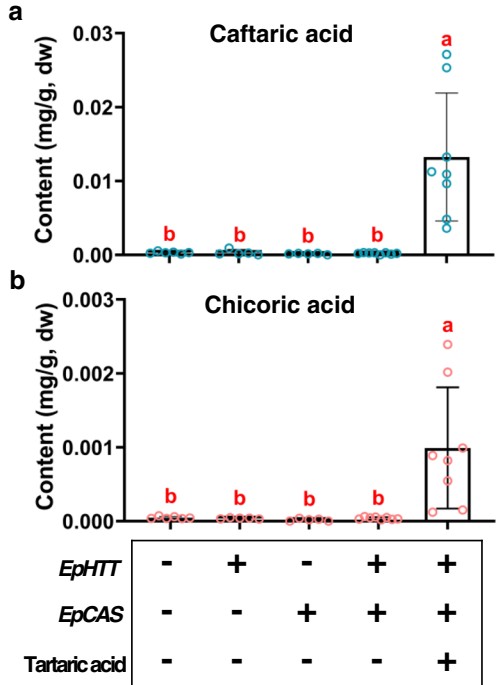

**a**

**Caftaric acid**

**b**

**Chicoric acid**

| EpHTT | - | + | - | + | + |
| EpCAS | - | - | + | + | + |
| Tartaric acid | - | - | - | - | + |

**Fig. 6 Reconstruction of chicoric acid biosynthesis in *Nicotiana benthamiana* leaves. a** Caftaric acid and **b** chicoric acid contents of *N. benthamiana* leaves infiltrated with different combinations of genes and precursor. Data are mean±s.d. ($n$ = 8, 5, 5, 6, 8 biologically independent samples, respectively). Lowercase letters indicate significant differences from each other ($P < 0.05$) analyzed by one-way ANOVA with Tukey's multiple comparisons test. Source data are provided as a Source Data file.

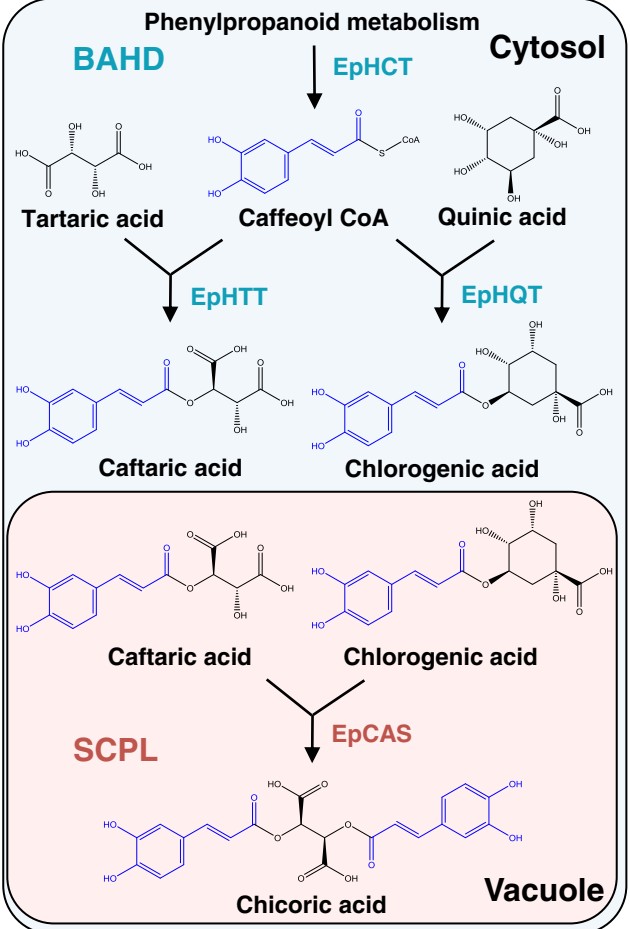

**Fig. 7 Schematic representation of the proposed chicoric acid biosynthetic pathway in purple coneflower.** Two types of acyltransferases distributed in distinct subcellular compartments are involved in the biosynthesis of chicoric acid. In the cytosol, the BAHDs including EpHTT and EpHQT use caffeoyl CoA from phenylpropanoid metabolism as an acyl donor to synthesize caftaric acid and chlorogenic acid, respectively. Both products are then transported into the vacuole. EpCAS, a specialized SCPL acyltransferase, uses chlorogenic acid as its acyl donor instead of 1-*O*-caffeoyl-β-D-glucose and transfers the caffeoyl group to caftaric acid to generate chicoric acid.

purple coneflower, EpCAS utilizes chlorogenic acid rather than 1-*O*-caffeoyl β-glucose as its acyl donor[13,37]. Among the species we tested, this pathway is unique, existing only *Echinacea* species and it is likely that chicoric acid biosynthesis is polyphyletic and has evolved by convergent mechanisms in other plant lineages. Chicoric acid is also vacuole-localized in *Cichorium endivia*[48] where it may be synthesized using caffeoyl CoA as an acyl donor as suggested for *Equisetum arvense*[7], or produced by modification of other intermediates, such as *p*-coumaroylcaffeoyl tartaric acid. Further research is needed to clarify its biosynthesis in other species. Additionally, it will be interesting to identify the biological functions (if any) of chicoric acid in *Echinacea* and other species. This could help us better understand its evolution and regulation. In summary, our results expand our understanding of acyltransferases in specialized metabolism and highlight the rapid evolution possible with these types of enzymes.

## Methods

**Plant materials**. Seeds of *Echinacea purpurea* (No. PI656830) were obtained from United States National Plant Germplasm System. Seeds were pretreated with 40 °C water at for 6 h and then grown in sterilized soil in the greenhouse (23 ± 2 °C, 16 h light/8 h dark). Two-month-old seedings were used for profiling of caffeic acid derivatives. Germination of sterile seedings was conducted as follows: seeds were sterilized with 5% Plant Preservative Mixture (PPM) (Plant Cell Technology, USA) in 3× Murashige and Skoog (MS) basal salt medium for 10 h and washed with sterile water three times. Superfluous water was absorbed with sterile filter paper. The seeds were plated on MS solid medium in the dark (covered with aluminum foil paper) for 2 weeks at 23 °C before germination. Seedings were transferred into an incubator (23 °C, 16 h light/8 h dark) and cultivated for one month before hairy root production. *Nicotiana benthamiana* was grown in sterilized soil in a greenhouse (23 ± 2 °C, 16 h light/8 h dark) for subcellular localization and metabolic engineering. *Helianthus annuus*, *Chrysanthemum coronarium*, *Cichorium endivia*, and *Lactuca sativa* were collected from a local supermarket.

**Extraction of caffeic acid derivatives**. Plant materials were collected and frozen in liquid nitrogen immediately and stored at -80 °C. On the day of extraction, samples were lyophilized and ground into powder. Ten milligrams dry sample were weighed and extracted with 1 mL 70% methanol (v/v) in an ultrasonic bath (Qsonica 700, USA) with 50% amplitude at 4 °C, 20 s on 40 s off, for a total of 15 min. Extracts were centrifuged at 5,000 × $g$ for 10 min at 4 °C and the supernatants were collected. The residue was extracted again using the same conditions. The two supernatants were combined and diluted to 2.5 mg/mL (dry material vs. solvent) for analysis using an ultra-performance liquid chromatography-diode array detector (UPLC-DAD) for detection of caffeic acid derivatives (CADs) and 0.5 mg/mL for liquid chromatography-high-resolution mass spectrometry (LC-HRMS) analysis. The extracts were further centrifuged at 20,000 × $g$ for 10 min at 4 °C and the supernatants were collected and stored at -80 °C before analysis.

**Qualitative analysis of CADs using LC-HRMS system**. CADs were qualified based on the comparison of their retention times and mass spectra from samples and standards using LC-HRMS (Nexera UHPLC LC-30A and AB SCIEX qTOF X500R) coupled with a Turbol V™ source and SCIEX OS software (version 1.7). The LC analytical conditions were as follows: samples were separated using a Hypersil Gold C18 column (100 × 2.1 mm, 1.9 μm; Thermo Fisher Scientific, USA) and the column temperature was set at 40 °C. The flow rate was 0.4 mL/min. The mobile phases were 0.1% formic acid (A) and acetonitrile (B), and the gradient was as follows: 2% B for 0.5 min; 2–20% B for 5.5 min; 20–95% B for 4 min; 95% B for 2 min followed by a decrease to 2% B for 0.1 min and re-equilibration of the

column for 2.9 min with 2% B. The injection volume was 1.0 µL and the sampler temperature was set at 15 °C. Mass spectrometry was performed using an electrospray ionization (ESI) source operating with the information-dependent acquisition (IDA) model. The source parameters in negative polarity were as follows: ion source gas 1: 50 psi; ion source gas 2: 50 psi; curtain gas: 35 psi; CAD gas: 7; temperature: 450 °C; spray voltage: −4500 V; for TOF-MS, the mass range: 100–1000 Da; declustering potential (DP): −80 V; DP spread: 0; collision energy (CE): −10 V; CE spread: 0; accumulation time: 0.1 s; IDA criteria: small molecule; for TOF-MS/MS, the mass range: 50 https://mpsde3.mpslimited.com/Digicore/DigiEditPage.aspx?FileName=421352456100491411330137.xml 1000 Da; DP: −80 V; DP spread: 0; CE: −40 V; CE spread: 20 V; accumulation time: 0.05 s.

**Quantitative analysis of CADs using UPLC-DAD system.** CAD contents were determined quantitatively using a UPLC-DAD system. Three CAD standards including caftaric acid, chicoric acid and chlorogenic acid were used. Retention times and the UV–vis spectra of standards and samples were compared for confirmation. Quantification was performed on a DIONEX UltiMate 3000 UHPLC system with Chromeleon software (Thermo Fisher Scientific, USA). Sample separation was performed using a Hypersil Gold C18 column (100 × 2.1 mm, 1.9 µm; Thermo Fisher Scientific, USA), and the column temperature was set at 40 °C. The mobile phases were 0.1% formic acid water (A) and acetonitrile (B). The gradient was as follows: 2% B for 0.5 min; 2–14% B for 1 min; 14–18% B for 5.5 min; 18–80% B for 2 min; 80% B for 1 min, followed by re-equilibration of the column for 3 min with 2% B. The flow rate was set as 0.5 mL/min and the injection volume was 2.0 µL. The sampler temperature was 10 °C. UV–vis absorption spectra were recorded on-line from 190 to 800 nm. Detection was conducted at 330 nm for quantitative purposes. Quantifications were carried out using external standards for calibration curves. The calibration curve coefficients for caftaric acid, chicoric acid, and chlorogenic acid were all above 0.9999. The results were expressed as mg of each CADs per 1 g dry material.

**Generation of _Echinacea purpurea_ hairy roots.** The hairy roots were constructed as previously reported[49]. Briefly, single colonies of pK7WG2R[50] plasmid containing _Agrobacterium rhizogenes_ A4 strain were inoculated into 5 mL of tryptone yeast broth (with 10 mM CaCl₂, 50 mg/L spectinomycin, and 50 mg/L kanamycin). The cultures were incubated at 28 °C overnight with shaking (180 rpm). The cultures were centrifuged for 15 min at 3000 × _g_ at 4 °C and resuspended in MS liquid medium containing 200 µM acetosyringone to make infection solution.

Leaf explants from one-month-old _Echinacea purpurea_ plants grown on MS medium were scratched by a sterilized knife blade in cold, sterilized water. Explants were immersed in infection solution for 5 min and blotted dry on sterile filter paper. Infected explants were then cocultured on MS medium containing 200 µM acetosyringone at 25 °C in the dark for 3 days before moving into MS medium supplemented with cefotaxime (250 mg/L). Hairy roots developed on cut ends of tissue were cultured in fresh 1/2 B5 solid medium (with 250 mg/L cefotaxime) at 25 °C in the dark. Hairy root cultures were transferred to fresh solid medium every 3 weeks and maintained as separate independent clones. Elongated root tips were transferred to flasks containing 30 mL of 1/2 B5 liquid medium (250 mg/L cefotaxime) to make liquid cultures, which were subcultured every week.

**Methyl jasmonate treatment and RNA-sequencing.** Empty vector-induced hairy roots were used for methyl jasmonate (MeJA) treatment. Hairy root cultures coming from independent progenitor roots were prepared and part of each independent culture received 100 µM MeJA and the other part was used as the control. The hairy roots were collected after 24 h for RNA-sequencing and after 48 h for profiling of CADs[51].

Total RNA was extracted using an RNeasy Plant Mini Kit (Qiagen, German). RNA-sequencing was performed using the BGI Illumina-HiSeq2000 platform. Library construction, sequencing, and analysis were carried out by the Beijing Genomic Institute (Shenzhen, China).

**Crude protein extracts.** The plant material was frozen in liquid nitrogen and ground into a fine powder. One gram of powdered material was extracted with 4 mL extraction buffer (phosphate buffer saline (PBS: 137 mM NaCl, 2.7 mM KCl, 10 mM Na₂HPO₄, and 2 mM KH₂PO₄) pH 7.0 containing 10 mM EDTA, 2% PVPP, 5 mM DTT, and 1× proteinase inhibitor cocktail (product No.: 4693132001, Roche) on ice for 2 h. The mixture was centrifuged at 4 °C at 5,000 × _g_ for 10 min. The supernatant was further centrifuged at 4 °C at 15,000 × _g_ for 10 min. The supernatant was filtered through a 0.22 µm polyethersulfone (PES) membrane and dialyzed against PBS buffer (pH 7.0) overnight at 4 °C refrigeration to remove low molecular weight compounds. The protein concentration was determined using a BCA kit (Sangon Biotech, China).

**Assay conditions for BAHD enzymes during protein purification.** A 100 µL reaction containing PBS buffer (pH 7.0), 20 µL crude protein extract (0.52 µg/µL), 120 µM caffeoyl CoA, and 120 µM tartaric acid or quinic acid was used for EpHTT and EpHQT assays, respectively. The mixture was incubated at 30 °C for 30 min and the reaction was stopped by adding 300 µL methanol. The supernatant was collected after centrifugation at 20,000 _g_ at 4 °C for 10 min and 2.0 µL was injected

for UPLC-DAD analysis. The UPLC conditions were the same as for quantitative analysis of CADs except for the gradient which was isocratic 5% B for 10 min.

**Purification of BAHD enzymes.** First, ammonium sulfate precipitation was used to fractionate crude protein extracts. Following pre-tests, 0–30%, 30–40%, 40–60%, and >60% (NH₄)₂SO₄ fractions were collected. After the addition of (NH₄)₂SO₄ powder, the mixture was shaken at 4 °C for 1 h and then centrifuged at 20,000 _g_ at 4 °C for 15 min. The supernatant was used for further precipitation and the sediment was resuspended in PBS (pH 7.0). (NH₄)₂SO₄ was removed by dialysis against PBS (pH 7.0) at 4 °C overnight. During this period, PBS buffer was replaced three times. The enzymes were purified further by FPLC (Fast Protein Liquid Chromatography, ÄKTA, GE Healthcare) coupled with HiTrap Q HP column (GE Healthcare) using a gradient of increasing concentration of NaCl from 0 to 1 M in PBS buffer (pH 7.0). The enzyme activities of fractions were measured, and those fractions with the greatest activity were loaded into SDS-PAGE gels and run for 5 min. When all the samples had entered the separation gel, each lane was cut out and processed for protein MS identification as detailed below.

**Protein MS identification.** Gels segments cut from SDS-PAGE gels were processed for protein MS identification. Peptides were digested by trypsin (Promega) and analyzed using nanoLC-QTOF-MS/MS (Ekspert NanoLC 425 and AB SCIEX Triple TOF 5600[52]. Peptides were separated using a SiELC ChromXP C18 column (75 µm × 15 cm, 3 µm, 120 Å). LC flow rate was set to 0.3 µL/min with phase A (2% acetonitrile, 98% water, 0.1% formic acid) and phase B (98% acetonitrile, 2% water, 0.1% formic acid). Gradients were: 5–7% B for 1 min, 7–22% B for 31 min, 32–38% B for 10 min, 32–55% B for 7 min, 55–80% B for 0.5 min, and hold at 80% B for 4.5 min, followed by re-equilibration with 5% B for 10 min. The MS method was as follows: TOF MS: CUR: 30; GS1: 12; GS2: 0; IHT: 150; ISVF: 2300; Mass Range: 300–1250 Da; Accumulation Time: 250 ms. IDA Criteria: 20; TOF MS/MS spectra acquired after each TOF MS scan; Rolling CE was applied. Mass Range: 100–1500 Da; Accumulation Time: 50 ms; CES: 5. Peptides were analyzed with ProteinPilot software and compared with the transcriptome database.

**Expression of recombinant BAHD enzymes in _E. coli_ and purification.** Full-length CDS of _EpHTT_, _EpHQT_, and _EpHCT_ were amplified from _Echinacea purpurea_ hairy root cDNA using primers listed in Supplementary Table 4 and cloned into the pDEST17 plasmid with 6 x His at the N-terminus[53]. The recombinant plasmids were transformed into _E. coli_ BL21(DE3) which was grown on LB solid medium (50 µg/mL ampicillin). Overnight cultures were transferred to 200 mL LB liquid medium containing antibiotics until the OD₆₀₀ reached 0.5–1.0. Isopropyl β-D-1-thiogalactopyranoside was added to a final concentration of 0.4 mM, and the cultures were grown overnight at 16 °C with shaking at 200 rpm. Cells were harvested by centrifugation at 5,000 × _g_ for 5 min.

Cells were resuspended in lysis buffer (25 mM Tris–HCl 8.0, 150 mM NaCl, 0.5 mM tris(2-carboxyethyl)phosphine). ATP (1 mM) and PMSF (1 mM) were also added. The cells were broken using a disruptor with 600 bar. After centrifugation at 20,000 × _g_ at 4 °C for 30 min, the supernatant was collected. Protein was purified using a Ni²⁺-NTA column (Qiagen) according to the manufacturer's instructions. Imidazole was removed using FPLC coupled with HiTrap Q HP column for purification of enzymes from crude extracts. The protein concentration was determined using a BCA kit (Sangon Biotech, China). Recombinant proteins were stored in PBS (pH7.0 with 10% glycerol) at -80 °C before use.

**Recombinant BAHD activities and kinetics.** According to our preliminary results, a UPLC based endpoint method was adopted for the determination of kinetics. Standard BAHD in vitro enzyme assays were performed in a 200 µL reaction mix containing PBS buffer (at appropriate pH), acyl-CoA thioesters as the acyl donors, and organic acids as the acyl acceptors. Purified protein (500 ng) was added to start the reaction. The mixture was incubated at 30 °C for 30 min. Methanol (300 µL) was added to stop the reaction. After centrifugation at 20,000 × _g_ at 4 °C for 10 min, the supernatant was collected and analyzed quantitatively using UPLC-DAD under the same conditions used for purification of BAHD activity. The enzyme reaction products were identified using LC-HRMS under the same conditions used for profiling of CADs. Enzyme activity was calculated as production of products and expressed as pkat (pmol/s). Kinetic parameters were determined from Michaelis–Menten plots using GraphPad Prism software (version 8.01). All in vitro enzyme assays were repeated at least three times.

For investigation of the effects of pH on BAHD activity at different pH values (from 2 to 9), PBS buffers were used in the enzyme assays. 100 µM tartaric acid or quinic acid and 100 µM caffeoyl CoA were used in the enzyme assays. The generation of products was quantified using UPLC.

For kinetic analyses, PBS pH 7.0 was selected as an optimum buffer. The kinetics for the acyl donors of EpHTT were determined using 0.5 mM tartaric acid and caffeoyl CoA (6.25–62.5 µM), p-coumaroyl CoA (12.5–500 µM), feruloyl CoA (25–200 µM), respectively. When determining the kinetics for acyl acceptors for EpHTT, different combinations were used, including caffeoyl CoA (50 µM) and tartaric acid (200–2400 µM), p-coumaroyl CoA (100 µM) and tartaric acid (500–4000 µM), and feruloyl CoA (100 µM) and tartaric acid (250–4000 µM). Quinic acid was fixed at 1 mM and various acyl-CoA thioesters (caffeoyl CoA

(6.25–62.5 μM); *p*-coumaroyl CoA (25–800 μM) and feruloyl CoA (50–400 μM)) were used for the study of the kinetics of EpHQT activity with different acyl donors. The kinetics of EpHQT's activity with different acyl donors were measured using the following combinations: caffeoyl CoA (50 μM) and quinic acid (200–2400 μM), *p*-coumaroyl CoA (100 μM) and quinic acid (250–2000 μM), and feruloyl CoA (100 μM) and quinic acid (1000–5000 μM). Products including *p*-coumaroylquinic acid, *p*-coumaroyltartaric acid, feruloylquinic acid, and feruloyltartaric acid were identified using LC-HRMS under the same conditions described above. Quantification was based on UPLC-DAD using caftaric acid and chlorogenic acid as references for tartaric acid esters and quinic acid esters, respectively.

For investigating the substrate preferences of BAHDs, acyl-CoA thioesters (100 μM) and organic acids (2 mM) including tartaric acid, quinic acid, and shikimic acid were used. The products were measured with LC-HRMS.

**Assay conditions for SCPL enzyme during protein purification**. Optimum conditions for SCPL activity were established by pre-tests. Assays were performed in a 200 μL in vitro enzyme reaction containing chlorogenic acid (250 μM), caftaric acid (250 μM), 10 mM $Na_2HPO_4$, and 2 mM $KH_2PO_4$ (pH 4.0). A total of 20 μL crude protein extract (0.4 μg/μL) was added to start the reaction. After incubation at 30 °C for 1 h, the reaction was stopped by adding 300 μL methanol. The mixture was centrifuged at 4 °C at 20,000 × *g* for 10 min and the supernatant was used for UPLC analysis under the same conditions as described above except the gradients were as follows: 2% B for 0.5 min, 2–15% B for 3.5 min, stay 15% B for 3 min, then drop to 2% B in 0.1 min and re-equilibrated with 2% for 2.9 min.

**Purification of chicoric acid synthase activity**. Similar to the purification of BAHD enzymes, ammonium sulfate precipitation was used to separate chicoric acid synthase activity initially. Following pre-tests, 0–20%, 20–40%, 40–60%, and >60% $(NH_4)_2SO_4$ fractions were used for fractionation. The fractions were prepared and desalted as described for BAHD acyl transferases. The active fraction was separated further by two steps including the use of the same anion exchange column used for BAHD purification and sieve chromatography. Molecular sieve separation was performed on FPLC with Superdex 200 10/300 GL (GE Healthcare) using PBS buffer (pH 7.0). Finally, the active fraction was used for the identification of EpCAS using the protein MS identification method described above.

**Recombinant SCPL enzyme expression in *S. cerevisiae* and purification**. The full-length cDNA of *EpCAS* was cloned into the pESC-His expression vector[54] with a 6×His tag at the C-terminus using ClonExpress II One Step Cloning Kit (Vazyme). *Saccharomyces cerevisiae* BY4741 was used as the host yeast strain. Yeast transformation procedures were conducted using a lithium acetate method[55] and were incubated at 30 °C. Single colonies were grown on SD dropout media containing 2% dextrose without Histidine (Solarbio) and were picked and then used for purification by inoculation into 15 mL of SD dropout medium containing 2% dextrose. Cells were grown overnight at 30 °C on a rotary shaker set at 200 rpm. Cultures were then inoculated in 200 mL of SD dropout media containing 2% dextrose and grown to an $OD_{600}$ of 0.4. All cells were transferred into 200 mL SD dropout medium containing 2% galactose. Yeast cells were grown at 30 °C on a rotary shaker for 36 h. The cells were centrifuged at 1500 × *g* for 5 min at 4 °C. The supernatant was discarded, and the cells were resuspended in 10 mL of sterile water. Cells were centrifuged at 1500 × *g* for 5 min at 4 °C. The supernatant was discarded, and the cell pellets were stored at -80 °C for later experiments.

Cells were resuspended in the same volume of Ni-NTA lysis buffer (50 mM $NaH_2PO_4$, 300 mM NaCl and 10 mM imidazole, pH 8.0) and broken using acid-washed glass beads (425–600 μm, Sigma) by shaking in a grinding machine (60 Hz, 60 s for 5 times) and then centrifuged at 5,000 × *g* at 4 °C for 10 min. The supernatant was collected and further centrifuged at 20,000 × *g* at 4 °C for 10 min. After passage through a 0.22 μm PES membrane, the protein was purified using a $Ni^{2+}$-NTA column (Qiagen) according to the manufacturer's instructions with some modifications. The column was washed with 20 mM imidazole and eluted using pH 4.5 buffer (50 mM $NaH_2PO_4$, 300 mM NaCl). Protein concentration was determined using a BCA kit (Sangon Biotech, China) and recombinant protein was stored at -80 °C before use.

**Mutagenesis and expression of EpCAS in *S. cerevisiae***. Overlapping gene fragments were separately amplified using *EpCAS* as a template with primers containing the desired single and combined amino acid substitutions[56]. The fragments were simultaneously cloned into pESC-His vector for expression as described above.

**Determination of recombinant EpCAS characteristics**. Standard conditions for EpCAS assays were 100 μL reactions containing chlorogenic acid (200 μM), caftaric acid (100 μM), 10 mM $Na_2HPO_4$, and 2 mM $KH_2PO_3$ (pH 4.0). Protein (1.41 μg) was added to start the reaction. After incubation at 30 °C for 12 h (based on pre-experiments), the reaction was stopped by adding 400 μL methanol. The mixture was centrifuged at 4 °C at 20,000 × *g* for 10 min and the supernatant was used for UPLC-DAD analysis under the same conditions used for the SCPL enzyme assay

during purification. Reactions were also analyzed using LC-HRMS by the same method as used for assaying BAHD activities.

For determination of the effects of pH, starting solution (10 mM $Na_2HPO_4$ and 2 mM $KH_2PO_3$) was adjusted into different pH (ranging from pH 1.6 to 10.7) by adding 1 M HCl or 1 M NaOH, and all other procedures were performed using standard conditions. For further identification of EpCAS as an SCPL protein, PMSF at different concentrations including 1, 5, and 10 mM was used to inhibit enzyme activity. For product and substrate degradation assays, 100 μM chicoric acid, 100 μM caftaric acid and 200 μM chlorogenic acid were used. To establish whether caftaric acid and caffeic acid can generate chicoric acid, 100 μM caftaric acid and 200 μM caffeic acid were combined. For kinetic analysis, caftaric acid ranging from 10 to 200 μM was mixed with 200 μM chlorogenic acid and chlorogenic acid ranging from 20 to 200 μM was mixed with 200 μM caftaric acid and assays were incubated with EpCAS for caftaric acid and chlorogenic acid analysis, respectively. For kinetic analysis against 1-*O*-caffeoyl β-D-glucose, caftaric acid 200 μM was mixed with 1-*O*-caffeoyl β-D-glucose ranging from 10 to 400 μM. For the characterization of activities with other acyl donors, 100 μM caftaric acid was mixed with 200 μM candidate donors including chlorogenic acid, 1-*O*-caffeoyl β-D-glucose, neochlorogenic acid, cryptochlorogenic acid and 5-*O*-caffeoylshikimic acid. The products were determined using LC-HRMS and UPLC-DAD. All in vitro enzyme assays were conducted at least three times.

**Synthesis of 1-*O*-caffeoyl-β-D-glucose**. 1-*O*-caffeoyl-β-D-glucose was synthesized according to published methods[57,58], and its identity was confirmed using LC-MS and NMR. UV/Vis: $\lambda_{max}$ 330 nm. [1]H NMR (400 MHz, $CD_3OD$) δ:7.66 (d, J = 16.0 Hz, 1H), 7.06 (s, 1H), 6.97 (d, J = 8.4 Hz, 1H), 6.79 (d, J = 8.0 Hz, 1H), 6.31 (d, J = 16.0 Hz, 1H), 5.58 (d, J = 7.6 Hz, 1H), 3.86 (d, J = 12.0 Hz, 1H), 3.72–3.68 (m, 1 H), 3.47–3.39 (m, 4H). HRMS (m/z): [M]⁻ calculated. for $C_{15}H_{18}O_9$, 341.0878; found, 341.0881 (Supplementary Fig. 12).

**Phylogenetic analysis**. The sequences of BAHD (CcsHCT (AFL93686), CcsHQT1 (CAM84302) and CcsHQT2 (CAR92145) from *Cynara cardunculus var. scolymus*; CiHCT1 (ANN12608), CiHCT2 (ANN12609), CiHQT1 (ANN12610), CiHQT2 (ANN12611), and CiHQT3 (ANN12612) from *Cichorium intybus*; NtHCT (CAD47830) and NtHQT (CAE46932) from *Nicotiana tabacum*; CaHCT (CAJ40778) from *Coffea arabica*; CcHQT (ABO77957) from *Coffea canephora*; TpHCT1A (ACI16630) from *Trifolium pratense*; AtHCT (AED95744) from *Arabidopsis thaliana*; SbHCT (4KE4_A) from *Sorghum bicolor*; AsHHT1 (BAC78633) from *Avena sativa*; CcaHQT (ABK79690) from *Cynara cardunculus var. altilis*; SlHQT (CAE46933) from *Solanum lycopersicum*; MpHCT (AXN55971) from *Marchantia paleacea*; PaHCT (AXN55972) from *Plagiochasma appendiculatum*; CsHCT (AEJ88365) from *Cucumis sativus*; PpHCT1 (AMK38063) from *Physcomitrium patens*) and SCPL (SpGAC (AAF64227) from *Solanum pennellii*; AtSMT (AAF78760), AtSCT (AAK52316), AtSAT (AEC07395), AtSST (AEC07397) and AtSCPL17 (AAS99709) from *Arabidopsis thaliana*; BnSCT1 (AAQ91191) and BnSCT2 (CAM91991) from *Brassica napus*; DkSCPL1 (BAF56655) and DkSCPL2 (BAH89272) from *Diospyros kaki*; CtAT1 (BAF99695) from *Clitoria ternatea*; AsSCPL1 (ACT21078) from *Avena strigosa*; OsIAA-At (EEE95946) from *Oryza sativa*) were obtained from NCBI and other public databases. Amino acid sequences were aligned and the Neighbor-Joining (NJ) trees were built using MEGA6 (Molecular Evolutionary Genetics Analysis version 6.0) with the following parameters: at least 1000 bootstrap replications, Poisson model, uniform rates and complete deletion[59].

**Subcellular localization**. Target genes were cloned into pSuper1300-GFP[60] using primers listed in Supplementary Table 4 to create in-frame, C-terminal GFP fusion proteins, and transformed into *Agrobacterium tumefaciens* strain GV3101. Transformed clones were grown on 2 mL LB selective medium (50 mg/L kanamycin, 50 mg/L gentamicin, and 50 mg/L rifampin) for 24 h at 28 °C with shaking at 180 rpm. The cultures were transferred to 20 mL LB medium containing antibiotics and grown until the $OD_{600}$ reached 1.0. Cells were harvested and resuspended in infiltration buffer (10 mM MES pH 5.6, 10 mM $MgCl_2$, and 200 μM acetosyringone) and incubated for an additional 2 h in the dark at room temperature. Strains containing target plasmids were infiltrated into the leaves of 4-week-old *Nicotiana benthamiana* plants. The plants were grown in the dark for 24 h (*EpHTT* and *EpHQT*) or 72 h (*EpCAS*) and prepared for microscopic observations. Leaves were cut into small pieces and placed on glass slides and observed with a ZEISS Cell Observer SD Spinning Disk Confocal Microscope. The vacuoles were stained by 10 μM SNARF-1 for 2 h before observation[61]. Detection parameters were as follows: GFP excitation at 488 nm and emission at 500–530 nm, SNARF-1 excitation at 514 nm and emission at 600–700 nm.

**Molecular docking**. The protein 3D structures of BAHD enzymes were modeled using I-TASSER (https://zhanglab.ccmb.med.umich.edu/I-TASSER/)[62] and CIS-RR[63]. Docking experiments were conducted on CB-Dock[64] based on AutoDock Vina[65]. EpCAS was modeled by SwissModel[66] using SERINE CARBOXYPEPTIDASE (PDB:IYSC) as template. AutoDock 4.2 was used for docking experiments[67]. Molecular graphics were rendered with PyMOL (version 2.3.2).

**Overexpression and RNAi of *BAHDs* and *SCPL* in hairy root cultures**. *EpHTT*, *EpHQT* and *EpCAS* genes were cloned into plasmid pK7WG2R with a 35 S promoter[50] using Gateway recombination according to the manufacturer's instructions for the overexpression of target genes in hairy roots. Nonhomologous sequences of *EpHTT*, *EpHQT* and *EpCAS* were amplified with the primers listed in Supplementary Table 4 and were cloned into binary hpRNA vector pK7WGIGW2R[68] using Gateway recombination for RNAi. The vectors were introduced into *Agrobacterium rhigogenes* A4 by electroporation. Successful clones were selected on tryptone yeast broth solid medium supplemented with specinomycin (50 mg/L) and kanamycin (50 mg/L). All other procedures for generation and analysis of hairy roots were as described as above.

**Quantitative reverse transcription polymerase chain reaction**. Total RNA was extracted with an RNeasy Plant Mini Kit (Qiagen). First-strand cDNA was synthesized from 1 μg RNA using a PrimeScript™ RT reagent Kit with gDNA Eraser (Takara). qRT-PCR was performed using a Bio-Rad CFX384 and iTaq Universal One-Step RT-qPCR Kits (Bio-Rad) according to the manufacturer's instructions. Results were calculated using Bio-Rad CFX manager software. Tubulin was used as internal control. The relative expression of genes was calculated using the ΔCt method. The primer pairs for qRT-PCR were designed by Primer3Plus (http://www.primer3plus.com) (Supplementary Table 4) and compared by BLAST analysis to the NCBI database for confirmation of primer specificity.

**Reconstruction of chicoric acid biosynthesis in *N. benthamiana***. For transient overexpression, *EpHTT* and *EpCAS* were cloned into the destination vector pEAQ-HT-DEST3[69] using Gateway recombination and transformed into *Agrobacterium tumefaciens* strain GV3101 by electroporation. Single clones with each target construct were transferred into 2 mL LB liquid culture (kanamycin, rifampicin, and gentamicin, 50 mg/L), and the cultures were grown at 28 °C in a shaker at 200 rpm for 1 day. For subculturing, 0.5 mL were transferred into 10 mL LB liquid medium (kanamycin, rifampicin, and gentamicin, 50 mg/L) and cells were grown for 6–8 h until the $OD_{600}$ reached 0.5–0.8. Cells were collected and resuspended in infiltration buffer (10 mM $MgCl_2$, 10 mM MES pH 5.6, and 250 μM acetosyringone) to $OD_{600} = 1.0$. The *Agrobacterium tumefaciens* stains with vectors carrying target genes were infiltrated into *Nicotiana benthamiana* leaves by syringe. Substrates (tartaric acid 1 mM, caftaric acid 500 μM) were injected into infiltrated leaves 5 days after inoculation. After 24 h, the leaves were collected for LC-HRMS analysis using the same conditions used for CAD profiling.

**Statistical analysis**. Unless specifically described, all the experiments in this paper were repeated at least three times and results from representative data sets are presented. GraphPad Prism (version 8.02) and Microsoft Excel (version Office 365) was used for the statistical analysis. The statistical evaluations used unpaired *t* tests and one-way analysis of variance (ANOVA) with multiple comparisons, followed by Tukey tests. The results were considered statistically significant at *$P < 0.05$.

**Reporting summary**. Further information on research design is available in the Nature Research Reporting Summary linked to this article.

## Data availability

Data supporting the findings of this work are available within the paper and its Supplementary Information files. A reporting summary for this Article is available as a Supplementary information file. The datasets and plant materials generated and analyzed during the current study are available from the corresponding author upon request. The sequences of the genes reported in this article have been deposited in NCBI GenBank: EpHTT (MT936803), EpHQT (MT936804), EpHCT (MT936805), and EpCAS (MT936806). The RNA-seq data from MeJA treatment and the control have been deposited in the SRA database of NCBI under accession numbers SRR8935731, SRR8935732, SRR8935733 for MeJA treatments and SRR8937034, SRR8937035, SRR8937036 for controls. The source data underlying Figs. 1d, 2d–g, 3c, 3d, 4a, 4b, and 6, Supplementary Figures 1, 3, 5, 9c, and 16a, as well as Supplementary Tables 1-3 are provided as a Source Data file.

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

## Acknowledgements

This work was supported by the National Natural Science Foundation of China (31670352 and 31800258), the Institutional Research Fund of Sichuan University (2020SCUNL106), the Fundamental Research Funds for the Central Universities (SCU2020D003), the China Postdoctoral Science Foundation (2018M631080) and the Post-Doctor Research Project, Sichuan University. C.M was supported by Institute Strategic Programs: Understanding and Exploiting Plant and Microbial Secondary 336 Metabolism (BB/J004596/1) and Molecules from Nature (BB/P012523/1) from the UK Biotechnology and Biological Sciences Research Council (BBSRC) to the John Innes Centre (JIC). We acknowledge the Mass Spectrometry Core Facility in College of Life Sciences, Sichuan University for the assistance in proteomics and metabolic analysis and Dr. Qing Zhao (Chenshan Botanical Garden, China) for advice on hairy root production and maintenance. We thank Prof. Xiaoya Chen (Institute of Plant Physiology and Ecology, Chinese Academy of Science), Prof. Jie Luo (Hainan University, China) and Prof. Chengchao Xu (Artemisinin Research Center, China) for critical comments on the manuscript.

## Author contributions

Y.Z. and C.M. devised the initial experimental strategies and R.F. and Y.Z. designed the experiments with C.M's advice. R.F. and P.Z. performed the experiments with the assistance of G.J. and L.W. Both S.Q. and Y.C. assisted with docking assays. R.F., C.M., and Y.Z. wrote the paper with inputs from all the authors who approved the final version.

## Competing interests

The authors declare no competing interests.
