## [Peer Review File · Nature Communications]

REVIEWER COMMENTS

Reviewer #1 (Remarks to the Author):

This manuscript presents the elucidation of the pathway for biosynthesis of chicoric acid in *Echinacea purpurea* and presumably other *echinacea* species. It is a very interesting story and presents a rather novel pathway that utilizes both BAHD transferases, as well as a serine carboxypeptidase-like protein acting as a hydroxycinnamoyl transferase. Although SCLPs have been implicated in these sorts of transferase reactions (e.g. SNG1 of *arabidopsis* that transfers sinapic acid from sinapoyl-glucose to malic acid), in the case of this chicoric acid biosynthetic pathway, the acyl donor appears to be the quinic acid moiety of chlorogenic acid, a quite novel finding.

This manuscript presents a lot of good work, is overall well written and will be of interest to the field. I do think there are a few issues that should be addressed. Although I make a lot of comments below, this is a really excellent body of work, and I recommend publication once these issues are addressed.

Lines 107-112: in talking about the HHLVD and HTLSD motifs, it would be useful to readers to give a citation for these.

Lines 159-160, the authors indicate that increased EpHQT expression increased chicoric acid content in hairy roots and refer to supplemental figure 8c. I'm confused by this because based on the figure, there's not a significant difference in chicoric acid accumulation in the overexpression lines. It does look like from the figure a decrease in HQT resulted in a significant decrease in chicoric acid accumulation. Is this what the authors meant?

Lines 171-175: These sentences are awkwardly written and need to be revised.

Lines 180-181: I think "stronger" activity is not quite the right way to put this. Maybe being more specific (catalytic efficiency??). That said, I think it can be very difficult to compare to data from a study done 25 years ago. Also, I've noticed different labs can carry out kinetic analyses quite differently, so again, comparisons between studies can be challenging.

Legend to supplementary figure 2 could use more detail. What is the coloration in panels a and b. What's the second line of sequence in panel c. Define the labels in panel c.

Line 209: The caffeic acid produced is a rather small amount and very hard to see in the figure. Perhaps final figure it is more apparent. That said, it might be worth indicating it is a small amount but is likely real since it's not seen with empty vector control.

Line 273: For this section, I think it would be helpful to the reader to indicate that this was done by infiltration of the gene constructs followed by infiltration of tartaric acid since *N. benthamiana* lacks that substrate.

Lines 296-297: It seems to me it might be appropriate to discuss the role of transport in this a bit more. How might these be transported to the vacuole and in what form (as a glycosylated derivative?). What enzymes might be involved in the transport. Does the chicoric acid stay in the vacuole? Where is chicoric acid localized in other species that perhaps use a different pathway?

Line 424: I think "PBS" needs to be better defined. I've seen phosphate buffered saline to specify a variety of formulations involving K and Na phosphate buffers with various amounts of NaCl and KCl present.

Line 432: can the authors indicate approximately how much protein corresponds to 20 ul of extract?

Line 348: "the gradient which was 5% B for 10 minutes" is confusing to me. Does this mean the starting point was different? Or do the authors mean the separation was done isocratically? Please clarify.

Lines 457-458: can you specified the column used here.

Line 503 on: I think more detail on how the rates were determined for the kinetic analyses should be provided. Did the authors depend the single 30 minute end point specified in the previous paragraph or did they sample and analyze the reactions over time. If a single end point was used, how do the authors know the rate determined at this single end point was a good estimation of the initial reaction rate? Additionally, what was the level of technical replication for these determinations? I know it's in a footnote in the supplemental tables, but I think it should be provided in the Methods section as well. Finally, it seems like some of the conditions used might not allow an accurate kinetic analysis, or at the very least, some caveats might apply to the values obtained. For example, as detailed in supplemental table 1, as the "fixed" substrate, tartaric acid was used at 500 uM, but the analyses to determine Km of tartaric acid indicate a Km equal to or greater than 500 uM. Consequently I think kcat would be underestimated in the determination of parameters for the CoA substrates and Km values might be inaccurate as well. Similarly, where the CoAs are the fixed substrates, some are not being used at a level much higher than their Km (thus, they aren't approaching being nonlimiting). There are similar issues for HQT determination in supplemental table 1. Fortunately, I don't think the kinetic parameters make or break the case for the pathway being proposed given some of the other experiments (e.g. the over expression, RNAi, reconstitution in *N. Benthamiana*), but these limitations to the kinetic analysis should perhaps be mentioned or justified.

Line 529: Are the crude protein extracts here the same ones described in line 422? If so, it might be helpful to the reader to indicate that.

Line 566: I presume cells were disrupted in a buffer. What was the buffer used for disruption.

Line 582 on: The reactions described use a relatively large amount of purified protein and a very long incubation time, so specific activity of the preparation seems pretty low. Is this worth mentioning in the results somewhere? Also, similar comment on the kinetic analyses as noted above.... How were the rates determined and are the authors confident the rates being measured are a good estimate of the initial rates of reaction.

Some minor points on grammar and word use:

Throughout methods: I don't think antibiotic and chemical names (e.g. Kanamycin, Gentamycin, Acetosyringone) should be capitalized.

Line 51: "use" rather than "uses".

Line 64: should end sentence with "family."

Line 180: I think "parameters" would be a more appropriate work instead of "characters".

Line 208: "result" rather than "resulted".

Line 216: delete "to"

Line 219: should read "minor amounts of caffeic acid."

Line 244: replace "tiny" with "low", "very low", or "(very) small amounts of"

Line 265: I suggest "protein extracts" instead of "proteins"

Line 270: "likely" instead of "like"

Line 287: should be "have been extensively"

Line 341: I think you can delete "two times"

Line 362: should there be units after the CAD gas value?

Line 637: should be "...acetosyringone) and incubated and additional...", I think.

Line 663: I suggest "All other procedures for generation and analysis of hairy roots were as described above."

Line 672: Authors should refer readers to the supplemental table that has the primers detailed.

Reviewer #2 (Remarks to the Author):

The authors have studied the biosynthesis of cichoric acid from Echinacea in great detail. In general, the methods used are adequate and state of the art.

The authors could identify a new enzyme (EPCAS) for cichoric acid biosynthesis and could localize it in the vacuole.

Overall, the ms is well written and informative.

However, a few items need consideration:

1. The discussion of convergent evolution for the cichoric acid synthase is not complete. The authors show that this protein is member of a family of similar proteins, albeit with a new but related function. This means that EPCAS is not a new invention but derived from an existing gene lineage. A true convergent evolution would be a gene from a completely different lineage. In the M&M and the figures, names for the species whose genes were compared should be provided.
2. The localisation analysis should be explained in more details and results should be shown in the main text. The traditional way to show such localisation would require the isolation of vacuoles and as a direct proof of vacuolar localisation.
3. For molecular docking, the authors should know that cichoric acid is a polyphenol; this means that the phenolic OH groups can dissociate under physiological conditions and form negative charged phenolate ions, which can form ionic bonds with charged amino groups of proteins. This issue needs to be taken into account and at least to be discussed.
4. It is true that Echinacea is an important medicinal plant. But there is no proof that cichoric acid should be the main active component. It is abundant but probably works with other polyphenols and alkylamides and other secondary metabolites. Thus some of the implications for this study over inflated.

5. There are several sloppy mistakes:

- Echinacea is not a family, but a genus!
- the names of taxonim authority is not written in italics
- names for chemicals are written without capitals (not Kanamycin)

6. Although the story is interesting for anybody working on the biosynthesis of secondary metabolites, the classical journal for such a study would be Phytochemistry or related journals.

Reviewer #3 (Remarks to the Author):

This study describes the identification and characterization of three acyltransferases involved in the biosynthesis of chicoric acid in Purple Coneflower (*Echinacea purpurea*). Two of these enzymes, EpHTT catalyzing the formation of caftaric acid, and EpHQT catalyzing the formation of chlorogenic acid, belong to the group of BAHD acyltransferases. They are located in the cytoplasm and use the coenzyme A thioester caffeoyl-CoA as activated acyl donor. Most interestingly, the third enzyme, EpCAS, belongs to the group of SCPL acyltransferases and is the first described member of this group using chlorogenic acid as acyl donor. Successful reconstitution of chicoric acid formation in tobacco leaves co-expressing EpHTT, EpHQT and EpCAS confirms that the authors have elucidated the chicoric acid biosynthesis pathway applied by *E. purpurea*.

The novelty of this study refers primarily to the final step of chicoric acid biosynthesis, the transfer of the caffeoyl moiety from chlorogenic acid to caftaric acid, and the catalyzing enzyme EpCAS, which have not been reported, so far. The results of this study are of interest to a wider field because chicoric acid is a bioactive compound used in pharmacy. By elucidation of biosynthesis, this article lays the foundation for future biotechnological production of this compound or breeding of cultivars with tailored chicoric acid content. In terms of research, this study improves our knowledge about the plasticity of acyltransferase function. With EpCAS, it adds a new unconventional member to the family of SCPL acyltransferases and produces new details about structure-function relationship and molecular evolution of this interesting group of enzymes. Remarkably, this study shows that solid biochemistry and enzymology remain of utmost importance for the identification of novel enzymes from non-model organisms, for which sequence- or coexpression-based prediction often fails.

In this study, the experiments employ an impressive broad spectrum of methods including enzyme purification, reverse cloning, hairy root transformation, heterologous protein production, enzyme kinetics, protein structure modeling and site-directed mutagenesis. They were carried out carefully, and the results produced justify the conclusions drawn by the authors.

There are some minor critical points I have to make:

- 1) Enzyme purification from plant material is a challenging approach. To reveal the efficiency of protein purification and the quality of the fractions used for peptide sequencing, it would be helpful to show the appropriate SDS-PAGES in Supplementary Fig. 1.
- 2) Heterologous production and purification of EpHTT, EpHQT and EpCAS should be illustrated by showing the SDS-PAGES of the purified enzymes in a new Supplementary Figure. This would allow for assessing the yield and the purity of the protein fractions used for kinetic measurements. Especially for EpCAS I consider this as necessary, because the heterologous production of active SCPL acyltransferases is challenging.
- 3) In the discussion part, lines 293-303 describe conclusions and should be organized in an extra paragraph.
- 4) There are three points that deserve more attention in the discussion paragraph. (1) pH optimum of EpCAS: A pH optimum of 4.0 appears unphysiologically, even for vacuolar enzymes. (2) Chicoric acid synthesis as described in this manuscript relies on transport of the precursors, caftaric acid and chlorogenic acid, from the cytoplasm into the vacuole. What is known about the transporters required? (3) Which alternative pathways could be applied by *Cichorium* and *Lactuca* to produce chicoric acid?
- 5) The text needs careful English editing. Starting with the very first word of the title, I detected a wealth of typing errors, which is in deep contrast to the scientific quality of the manuscript.

Dear Editor,

We would like to thank you and the reviewers for your helpful comments and suggestions for improving our manuscript. Below, we describe how we have addressed the comments made by the reviewers. We believe that our revisions have improved the manuscript substantially and we hope they meet your requirements.

Reviewer #1 (Remarks to the Author):

This manuscript presents the elucidation of the pathway for biosynthesis of chicoric acid in *Echinacea purpurea* and presumably other echinacea species. It is a very interesting story and presents a rather novel pathway that utilizes both BAHD transferases, as well as a serine carboxypeptidase-like protein acting as a hydroxycinnamoyl transferase. Although SCLPs have been implicated in these sorts of transferase reactions (e.g. SNG1 of arabidopsis that transfers sinapic acid from sinapoyl-glucose to malic acid), in the case of this chicoric acid biosynthetic pathway, the acyl donor appears to be the quinic acid moiety of chlorogenic acid, a quite novel finding.

This manuscript presents a lot of good work, is overall well written and will be of interest to the field. I do think there are a few issues that should be addressed. Although I make a lot of comments below, this is a really excellent body of work, and I recommend publication once these issues are addressed.

Response:

Thank you very much for these supportive comments of our work. We really appreciate all your valuable suggestions on our MS. We present point-to-point responses below.

Lines 107-112: in talking about the HHLVD and HTLSD motifs, it would be useful to readers to give a citation for these.

Response:

We have added more detailed descriptions and provided references (line 106-108).

Lines 159-160, the authors indicate that increased *EpHQT* expression increased chicoric acid content in hairy roots and refer to supplemental figure 8c. I'm confused by this because based on the figure, there's not a significant difference in chicoric acid accumulation in the overexpression lines. It does look like from the figure a decrease in *HQT* resulted in a significant decrease in chicoric acid accumulation. Is this what the authors meant?

Response:

Thank you very much for pointing out this discrepancy. We have corrected the description to: "decreases in *EpHQT* expression also led to the decline of chicoric acid content in transgenic hairy roots". (line 162-163)

Lines 171-175: These sentences are awkwardly written and need to be revised.

Response:

We have re-written these sentences (line 174-179).

Lines 180-181: I think “stronger” activity is not quite the right way to put this. Maybe being more specific (catalytic efficiency??). That said, I think it can be very difficult to compare to data from a study done 25 years ago. Also, I’ve noticed different labs can carry out kinetic analyses quite differently, so again, comparisons between studies can be challenging.

Response:

We agree with you that it is difficult to compare kinetics between different experiments undertaken by different researchers. We have deleted the direct comparison (line 186). In addition, we have removed the comparison of EpHTT activity with the results from 25 years ago (line 122-123).

Legend to supplementary figure 2 could use more detail. What is the coloration in panels a and b. What’s the second line of sequence in panel c. Define the labels in panel c.

Response:

Thank you for pointing out these problems. We have carefully updated the information. Firstly, we have increased the size of each panel and divided them into three separate supplementary figures (S2, 11 and 12). In panels a and b (now Supplementary Fig. 2 and 12), dark blue, hot pink, cyan and light yellow represent 100%, $\geq 75\%$, $\geq 50\%$ and $\geq 33\%$ amino acid identity, respectively. In panel c (now Supplementary Fig. 11), the second line of sequence was generated by the software showing S - the signal peptide amino acids, C - the cleavage site and X - other amino acids. The SP (Sec/SPI) represents the likelihood probability of a secretory signal peptide in EpCAS which is 0.9932. CS indicates the cleavage site between amino acids 25 and 26 of EpCAS. OTHER represents the probability that these amino acids in EpCAS do not have any kind of signal peptide, which is 0.0068. We have added these details to the figure legend (now Supplementary Fig.11).

Line 209: The caffeic acid produced is a rather small amount and very hard to see in the figure. Perhaps final figure it is more apparent. That said, it might be worth indicating it is a small amount but is likely real since it’s not seen with empty vector control.

Response:

In Figure 3b, the peak of caffeic acid is indeed quite small. We have added the XIC and MS2 of caffeic acid in Supplementary Fig. 10 f and g detected by LC-HRMS, which confirmed the existence of caffeic acid.

Line 273: For this section, I think it would be helpful to the reader to indicate that this was done by infiltration of the gene constructs followed by infiltration of tartaric acid since *N. benthamiana* lacks that substrate.

Response:

We have emphasized the lack of tartaric acid in *N. benthamiana* and the infiltration

order of genes and substrate and we revised the manuscript (line 288-290).

Lines 296-297: It seems to me it might be appropriate to discuss the role of transport in this a bit more. How might these be transported to the vacuole and in what form (as a glycosylated derivative?). What enzymes might be involved in the transport. Does the chicoric acid stay in the vacuole? Where is chicoric acid localized in other species that perhaps use a different pathway?

Response:

According to your and Reviewer #3's comments, we have added more detail to the discussion of these points.

The biosynthesis of chicoric acid depends on the transport of caftaric acid and chlorogenic acid to the vacuoles. Unfortunately, no transporter has yet been identified for hydroxycinnamic acid derivatives. Chlorogenic acid has been reported to be transported into vacuoles by vesicles in *Lonicera japonica* flowers¹. Based on this study, it is possible chlorogenic acid and caftaric acid are transported through the aggregation of vesicles in their unmodified forms. However, we cannot draw any firm conclusions due to the lack of direct data. A new MSc student has just started to study these transport mechanisms in the lab. To address this point further, we have made following changes: 1. For the model present in Fig 7. We have removed the original rectangles on the vacuole membrane to avoid any misunderstanding. 2. We specifically discuss the potential routes for transportation in the discussion (line 317-321). We believe by doing this, we have raised the question for future studies.

Chicoric acid in purple coneflower has been shown to be stored in vacuoles², in line with our proposed biosynthetic pathway. We have specifically added this into our MS (line 310-313).

For other chicoric acid containing species such as *Cichorium endivia*, chicoric acid has been assumed to be enriched in foliar parenchyma cell vacuoles³. We have added this into MS (Line 337-341)

Line 424: I think "PBS" needs to be better defined. I've seen phosphate buffered saline to specify a variety of formulations involving K and Na phosphate buffers with various amounts of NaCl and KCl present.

Response:

The description of the composition of PBS (137 mM NaCl, 2.7 mM KCl, 10 mM Na₂HPO₄ and 2 mM KH₂PO₄) has been added (line 455-456).

Line 432: can the authors indicate approximately how much protein corresponds to 20 ul of extract?

Response:

The concentration of crude protein was 0.52 mg/mL, and the approximate protein content of 20 µL of crude protein extract was 10.4 µg. We have added the concentration

(line 465-466).

Line 348: “the gradient which was 5% B for 10 minutes” is confusing to me. Does this mean the starting point was different? Or do the authors mean the separation was done isocratically? Please clarify.

Response:

The gradient was isocratic 5% B for ten mins. We have clarified this on line 471.

Lines 457-458: can you specified the column used here.

Response:

We have added the name of the column: SiELC ChromXP C18 column (75 μm ×15 cm, 3 μm , 120 Å) (line 492-493).

Line 503 on: I think more detail on how the rates were determined for the kinetic analyses should be provided. Did the authors depend the single 30 minute end point specified in the previous paragraph or did they sample and analyze the reactions over time. If a single end point was used, how do the authors know the rate determined at this single end point was a good estimation of the initial reaction rate? Additionally, what was the level of technical replication for these determinations? I know it's in a footnote in the supplemental tables, but I think it should be provided in the Methods section as well. Finally, it seems like some of the conditions used might not allow an accurate kinetic analysis, or at the very least, some caveats might apply to the values obtained. For example, as detailed in supplemental table 1, as the “fixed” substrate, tartaric acid was used at 500 μM , but the analyses to determine K_m of tartaric acid indicate a K_m equal to or greater than 500 μM . Consequently I think k_{cat} would be underestimated in the determination of parameters for the CoA substrates and K_m values might be inaccurate as well. Similarly, where the CoAs are the fixed substrates, some are not being used at a level much higher than their K_m (thus, they aren't approaching being nonlimiting). There are similar issues for HQT determination in supplemental table 1. Fortunately, I don't think the kinetic parameters make or break the case for the pathway being proposed given some of the other experiments (e.g. the over expression, RNAi, reconstitution in *N. Benthamiana*), but these limitations to the kinetic analysis should perhaps be mentioned or justified.

Response:

Thank you for these insightful comments. We have added more detailed descriptions of the methods used for kinetic determinations (line 526-527, 535-538).

We used a 30 min endpoint method for the determination of kinetic parameters. The reaction time of 30 min was based on our preliminary experiments and references in the literature⁴. We have attached the effects of reaction time on the yield of products below. An end point of 30 min was selected taking into account both the initial reaction rate and the accuracy of detection (a greater area ensures greater accuracy in calculating the integral).

The effects of reaction time on the yield of products

We have conducted all experiments at least three times to ensure repeatability. Details have been added in the methods section (line 535-538).

We understand concerns about the accuracy of kinetic analysis. All three enzymes use two substrates for their acyl transfer reactions. In the present study, due to limitations to the availability of CoA thioesters and the fact that high levels of organic acids inhibited enzyme activities through their effects on the pH of the reaction, we were able to measure only the apparent Michaelis constants. These results allowed us to compare the acyl donor preferences but not to compare these with other published work. Consequently, as suggested, we have deleted the comparison with other enzymes. Our data did allow us to conclude that caffeoyl CoA was preferred by EpHTT and EpHQT which coincidentally resulted the production of caftaric acid and chlorogenic acid.

Line 529: Are the crude protein extracts here the same ones described in line 422? If so, it might be helpful to the reader to indicate that.

Response:

Thank you for pointing out this confusion. The crude protein extract was different from BAHD purification, but with the similar material and the same extraction method. We have added the concentration of crude protein extract (line 572).

Line 566: I presume cells were disrupted in a buffer. What was the buffer used for disruption.

Response:

We have added: "Cells were resuspended in Ni-NTA lysis buffer (50 mM NaH₂PO₄, 300 mM NaCl and 10 mM imidazole, pH 8.0) and broken using acid-washed glass beads" (line 607-608).

Line 582 on: The reactions described use a relatively large amount of purified protein and a very long incubation time, so specific activity of the preparation seems pretty low. Is this worth mentioning in the results somewhere? Also, similar comment on the kinetic analyses as noted above.... How were the rates determined and are the authors confident the rates being measured are a good estimate of the initial rates of reaction.

Response:

We used 1.41 μg of recombinant EpCAS to perform the assays, which is almost triple the amount used for BAHDs (500 ng). We have added the SDS-PAGE of recombinant proteins in supplementary Fig.3 as suggested by Reviewer#3. The EpCAS band is fainter than those for BAHDs indicating a little lower purity. In addition, EpCAS showed quite low K_{cat} values against two substrates, only about 1% of HTT and 5% of HQT values, indicating a low catalytic speed. All these results demonstrated that the CAS reaction in the vacuole is slower than the HTT and HQT reactions in the cytosol. We have added related information in the MS (line 185-188).

The reaction time was optimized based on preliminary experiments.

The effects of reaction time on chicoric acid production by EpCAS

Several identified SCPLs in other species need relatively long reaction times, including SCPL1 from oats, where the assay for production of Avenacin A-1 was conducted at 30 °C for 24 hours⁵; AtSMT expressed in *E. coli* was incubated with substrates at 30 °C for 14 hours⁶. These results indicated the low catalytic speed of SCPL, in line with the idea that EpCAS is still evolving and represents a good example for the study of plant enzyme evolution.

We have added more discussion about it in the result part (line 184-188).

Some minor points on grammar and word use:

Throughout methods: I don't think antibiotic and chemical names (e.g. Kanamycin, Gentamycin, Acetosyringone) should be capitalized.

Response:

Corrected (line 681-682, 729-730, 732).

Line 51: "use" rather than "uses".

Response:

Corrected (line 52).

Line 64: should end sentence with “family.”

Response:

Corrected (line 64).

Line 180: I think “parameters” would be a more appropriate work instead of “characters”.

Response:

Corrected (line 186).

Line 208: “result” rather than “resulted”.

Response:

Corrected (line 213).

Line 216: delete “to”

Response:

Corrected (line 222).

Line 219: should read “minor amounts of caffeic acid.”

Response:

Corrected (line 226).

Line 244: replace “tiny” with “low”, “very low”, or “(very) small amounts of”

Response:

Corrected (line 251).

Line 265: I suggest “protein extracts” instead of “proteins”

Response:

Corrected (line 271).

Line 270: “likely” instead of “like”

Response:

Corrected (line 277).

Line 287: should be “have been extensively”

Response:

Corrected (line 299).

Line 341: I think you can delete “two times”

Response:

Corrected (line 371).

Line 362: should there be units after the CAD gas value?

Response:

We have checked the software and consulted the AB SCIEX application engineer and found that CAD gas value is just a level and has no unit (see below).

Line 637: should be "...acetosyringone) and incubated and additional...", I think.

Response:

Corrected (line 685).

Line 663: I suggest "All other procedures for generation and analysis of hairy roots were as described above."

Response:

Corrected (line 711-712).

Line 672: Authors should refer readers to the supplemental table that has the primers detailed.

Response:

Corrected (line 722).

In summary, thank you for your positive and helpful comments, which have helped us to improve our MS substantially.

Reviewer #2 (Remarks to the Author):

The authors have studied the biosynthesis of cichoric acid from Echinacea in great detail. In general, the methods used are adequate and state of the art.

The authors could identify a new enzyme (EPCAS) for cichoric acid biosynthesis and could localize it in the vacuole.

Overall, the ms is well written and informative.

Response:

Thank you very much for your positive comments.

However, a few items need consideration:

1. The discussion of convergent evolution for the cichoric acid synthase is not complete. The authors show that this protein is member of a family of similar proteins, albeit with a new but related function. This means that EPCAS is not a new invention but derived from an existing gene lineage. A true convergent evolution would be a gene from a completely different lineage. In the M&M and the figures, names for the species whose genes were compared should be provided.

Response:

Thank you very much for pointing out this confusion! In the present study, we did not compare identified proteins with corresponding homologous proteins from other species. our phylogenetic analyses failed to identify any sequences encoding proteins highly homologous to EpHTT and EpCAS in species from other genera that accumulate chicoric acid, although genes encoding HQT like proteins were found in these other species. These results were in line with the lack of HTT and CAS activities in crude protein extracts from these species. Since HTT and CAS activities were absent in *Cichorium endivia* and *Lactuca sativa*, we conclude that the biosynthesis of chicoric acid is polyphyletic and that the pathway, based on HTT and unique CAS activities in *Echinacea* is different to that operating in *Cichorium endivia* and *Lactuca sativa*. Our conclusion of convergent evolution was based on these differences in substrate and enzyme activities and not on the comparison of homologous proteins/genes, but is in line with the idea of “the same products from different substrates” proposed by Eran Pichersky and Efraim Lewinsohn⁷. We have revised this part to avoid any confusion (line 259-282).

2. The localisation analysis should be explained in more details and results should be shown in the main text. The traditional way to show such localisation would require the isolation of vacuoles and as a direct proof of vacuolar localisation.

Response:

We have moved the cellular localization pictures from supplementary figure 4 a and d to figure 2 h and 3 e and added better descriptions (line 117-119, 190-191).

Our present results provide enough evidence to support our conclusion about the vacuolar localization of EpCAS:

1) Our cellular localization data clearly show the vacuole-localization of EpCAS.

2) We have added a citation which reported that chicoric acid in purple coneflower is stored in vacuoles². This result is in line with our proposed chicoric acid biosynthetic pathway (line 310-312).

3. For molecular docking, the authors should know that chicoric acid is a polyphenol; this means that the phenolic OH groups can dissociate under physiological conditions and form negatively charged phenolate ions, which can form ionic bonds with charged amino groups of proteins. This issue needs to be taken into account and at least to be discussed.

Response:

Thank you for pointing this out. As suggested, we undertook a repeat analysis where negatively charged phenolate ions were introduced. We used AutoDock 4.2 software to conduct the molecular docking assay and adopted the rank#1 conformation to model the catalytic mechanism. Possible ionic bonds were taken into account. We obtained a similar result to before, which we have highlighted in Figure 4c. The coloration in Figure 4c has also been updated to avoid the same atoms having different colors. Several ionic bonds can be observed between OH groups from chicoric acid and NH₂ or NH from amino acids or peptide bonds, which would support the predicted conformation for the reaction.

4. It is true that Echinacea is an important medicinal plant. But there is no proof that chicoric acid should be the main active component. It is abundant but probably works with other polyphenols and alkylamides and other secondary metabolites. Thus some of the implications for this study over inflated.

Response:

We agree that there may be other components that contribute to the bioactivities of purple coneflower. Indeed, in the manuscript, we described chicoric acid as the “major bioactive compound”. Chicoric acid is an abundant metabolite in purple coneflower and is also used as an index for the quality of raw material and commercial products, such as GNC Echinacea product (GNC HERBAL PLUS® ECHINACEA EXTRACT 500 MG (<https://www.gnc.com/echinacea/198712.html>)), implying that chicoric acid is considered to be “the major active component” of purple coneflower.

The bioactivity of chicoric acid was reviewed last year⁸ and since then, at least 16 papers have been published on the bioactivities of chicoric acid. All these papers support the view that chicoric acid is an important natural product.

5. There are several sloppy mistakes:

- Echinacea is not a family, but a genus!
- the names of taxonomic authority is not written in italics
- names for chemicals are written without capitals (not Kanamycin)

Response:

Thank you! We have corrected the term family to genus (line 35). The Italics were changed to Roman (line 19) and the chemical names were corrected throughout the MS

(line 681-682, 729-730, 732).

6. Although the story is interesting for anybody working on the biosynthesis of secondary metabolites, the classical journal for such a study would be *Phytochemistry* or related journals.

Response:

In addition to the novelty of the pathway we elucidated, our study provides an excellent example for the methodology of pathway elucidation. We are delighted and grateful that both the handling editor and the reviewers considered that our manuscript would be of interest to readers of *Nature Communications*, a journal with increasing publications on natural plant products.

Special thanks to you for your good comments sincerely.

Reviewer #3 (Remarks to the Author):

This study describes the identification and characterization of three acyltransferases involved in the biosynthesis of chicoric acid in Purple Coneflower (*Echinacea purpurea*). Two of these enzymes, EpHTT catalyzing the formation of caftaric acid, and EpHQT catalyzing the formation of chlorogenic acid, belong to the group of BAHD acyltransferases. They are located in the cytoplasm and use the coenzyme A thioester caffeoyl-CoA as activated acyl donor. Most interestingly, the third enzyme, EpCAS, belongs to the group of SCPL acyltransferases and is the first described member of this group using chlorogenic acid as acyl donor. Successful reconstitution of chicoric acid formation in tobacco leaves co-expressing EpHTT, EpHQT and EpCAS confirms that the authors have elucidated the chicoric acid biosynthesis pathway applied by *E. purpurea*.

The novelty of this study refers primarily to the final step of chicoric acid biosynthesis, the transfer of the caffeoyl moiety from chlorogenic acid to caftaric acid, and the catalyzing enzyme EpCAS, which have not been reported, so far. The results of this study are of interest to a wider field because chicoric acid is a bioactive compound used in pharmacy. By elucidation of biosynthesis, this article lays the foundation for future biotechnological production of this compound or breeding of cultivars with tailored chicoric acid content. In terms of research, this study improves our knowledge about the plasticity of acyltransferase function. With EpCAS, it adds a new unconventional member to the family of SCPL acyltransferases and produces new details about structure-function relationship and molecular evolution of this interesting group of enzymes. Remarkably, this study shows that solid biochemistry and enzymology remain of utmost importance for the identification of novel enzymes from non-model organisms, for which sequence- or coexpression-based prediction often fails.

In this study, the experiments employ an impressive broad spectrum of methods including enzyme purification, reverse cloning, hairy root transformation, heterologous protein production, enzyme kinetics, protein structure modeling and site-directed mutagenesis. They were carried out carefully, and the results produced justify the conclusions drawn by the authors.

Response:

Thank you for your very positive comments.

There are some minor critical points I have to make:

1) Enzyme purification from plant material is a challenging approach. To reveal the efficiency of protein purification and the quality of the fractions used for peptide sequencing, it would be helpful to show the appropriate SDS-PAGES in Supplementary Fig. 1.

Response:

In the process of purification of BAHDs, HTT and HQT fractions with high activity following FPLC were loaded directly onto SDS-PAGE gels and run for only a few minutes until all proteins had entered the separating gel. We then stopped the

electrophoresis and cut out the corresponding lanes directly for peptide mass fingerprinting. We have added a more detailed description to methodology section (line 484-486).

During the purification of CAS activity, we had conducted SDS-PAGE for the fractions from FPLC separation steps, and we cut 50-100 kDa bands from fraction#13 after sieve-based purification for peptide mass fingerprinting. We have added the SDS-PAGE blots of CAS in supplementary figure 1d. The SDS-PAGE was conducted on Bio-Rad TGX Stain-Free FastCast Acrylamide kit and the blots were taken by a Bio-Rad ChemiDOC™ Touch instrument in stain-free model.

2) Heterologous production and purification of EpHTT, EpHQT and EpCAS should be illustrated by showing the SDS-PAGEs of the purified enzymes in a new Supplementary Figure. This would allow for assessing the yield and the purity of the protein fractions used for kinetic measurements. Especially for EpCAS I consider this as necessary, because the heterologous production of active SCPL acyltransferases is challenging.

Response:

Thank you for this valuable suggestion. We have provided these plots in supplementary figure 3. Panels a and b were taken by a digital camera after Coomassie blue staining and panel c was taken using a Bio-Rad ChemiDOC™ Touch instrument in stain-free mode.

In our present study, EpBAHDs were successfully expressed heterologously and purified by SDS-PAGE (Supplementary Figure 3 a and b). When we used Ni-NTA to purify EpCAS, using elution at pH 4.5 which showed higher enzyme activity than high concentration imidazole elution.

3) In the discussion part, lanes 293-303 describe conclusions and should be organized in an extra paragraph.

Response:

Thank you, we have revised this section (line 323-329).

4) There are three points that deserve more attention in the discussion paragraph. (1) pH optimum of EpCAS: A pH optimum of 4.0 appears unphysiologically, even for vacuolar enzymes. (2) Chicoric acid synthesis as described in this manuscript relies on transport of the precursors, caftaric acid and chlorogenic acid, from the cytoplasm into the vacuole. What is known about the transporters required? (3) Which alternative pathways could be applied by Cichorium and Lactuca to produce chicoric acid?

Response:

We have expanded the relevant discussions.

(1) For the optimum pH, we used *in vitro* enzyme assays to determine the pH optima of the different enzymes. The optimum for EpCAS of pH 4.0 was an *in vitro* result. A proposed SCPL, epicatechin:1-*O*-galloyl- β -D-glucose *O*-galloyltransferase (ECGT) showed maximal *in vitro* reaction rates between pH 4.0-6.0⁹. SIHQT also shows highest secondary enzyme activity at pH 4 when acting as a chlorogenate-

chlorogenate transferase to generate dicaffeoyl quinic acid in vacuoles¹⁰. A lower pH optimum likely ensures that EpCAS is active in vacuoles with more acidic pH (normally in the range of pH5-6). To avoid any misunderstanding, we have specifically empathized this in our revision (line 313-317).

- (2) For the discussion of transport of chlorogenic acid and caftaric acid to the vacuole please see our responses to Reviewer 1 (also see MS lines 317-321).
- (3) As we mentioned in our responses to reviewer 1, currently we can only say the substrate and enzyme activities in *Cichorium* and *Lactuca* are different to those of *Echinacea*. We have added related discussion about possible alternative biosynthetic pathways (line 334-341).

5) The text needs careful English editing. Starting with the very first word of the title, I detected a wealth of typing errors, which is in deep contrast to the scientific quality of the manuscript.

Response:

Our co-author, Prof. Cathie Martin, is a native speaker and has served for a long time as a senior editor for scientific journals. She has helped us to proof-read our revised MS thoroughly to ensure our use of English meets publication standards.

We appreciate your helpful comments, thank you very much.

Reference cited:

- 1 Li, Y. *et al.* Correlation of the temporal and spatial expression patterns of HQT with the biosynthesis and accumulation of chlorogenic acid in *Lonicera japonica* flowers. *Hortic Res* **6**, 73, doi:10.1038/s41438-019-0154-2 (2019).
- 2 Li, Z. *et al.* The Synthesis and Storage Sites of Phenolic Compounds in the Root and Rhizome of *Echinacea purpurea*. *Am J Plant Sci* **03**, 551-558, doi:10.4236/ajps.2012.34066 (2012).
- 3 Goupy, P. M., Varoquaux, P. J. A., Nicolas, J. J. & Macheix, J. J. Identification and localization of hydroxycinnamoyl and flavonol derivatives from endive (*Cichorium endivia* L. cv. Geante Maraichere) leaves. *J Agric Food Chem* **38**, 2116-2121, doi:10.1021/jf00102a003 (1990).
- 4 Niggeweg, R., Michael, A. J. & Martin, C. Engineering plants with increased levels of the antioxidant chlorogenic acid. *Nat Biotechnol* **22**, 746-754, doi:10.1038/nbt966 (2004).
- 5 Mugford, S. T. *et al.* A Serine Carboxypeptidase-Like Acyltransferase Is Required for Synthesis of Antimicrobial Compounds and Disease Resistance in Oats. *Plant Cell* **21**, 2473, doi:10.1105/tpc.109.065870 (2009).
- 6 Lehfeldt, C. *et al.* Cloning of the SNG1 gene of *Arabidopsis* reveals a role for a serine carboxypeptidase-like protein as an acyltransferase in secondary metabolism. *Plant Cell* **12**, 1295-1306, doi:10.1105/tpc.12.8.1295 (2000).
- 7 Pichersky, E. & Lewinsohn, E. Convergent evolution in plant specialized metabolism. *Annual review of plant biology* **62**, 549-566, doi:10.1146/annurev-arplant-042110-103814 (2011).

- 8 Peng, Y., Sun, Q. & Park, Y. The Bioactive Effects of Chicoric Acid As a Functional Food Ingredient. *J Med Food* **22**, 645-652, doi:10.1089/jmf.2018.0211 (2019).
- 9 Liu, Y. *et al.* Purification and characterization of a novel galloyltransferase involved in catechin galloylation in the tea plant (*Camellia sinensis*). *J Biol Chem* **287**, 44406-44417, doi:10.1074/jbc.M112.403071 (2012).
- 10 Moglia, A. *et al.* Dual catalytic activity of hydroxycinnamoyl-coenzyme A quinate transferase from tomato allows it to moonlight in the synthesis of both mono- and dicaffeoylquinic acids. *Plant Physiol* **166**, 1777-1787, doi:10.1104/pp.114.251371 (2014).

REVIEWERS' COMMENTS

Reviewer #1 (Remarks to the Author):

The authors have improved the manuscript with this revision and have addressed the main concerns I raised in my previous review. I am happy with the over all content of the manuscript, but do believe a few items still need to be addressed. I find the Methods section somewhat uneven, with sometimes too little detail given, and sometimes perhaps too much (for example where methods are relatively well established and could likely be conveyed via a citation). I'd suggest authors review that entire section to make sure appropriate level of detail is provided. I provide some examples below but it may not be exhaustive. There are also some places where the grammar and language use could be improved.

Specific comments:

Line 28: delete "be"

Lines 75-77: You can't really say this pathway only exists in Echinacea unless you examine every species! I'd suggest something like "... biosynthetic route has not been previously described and might exist only in Echinacea and it is likely...."

Line 175: I'd suggest replacing "traced" with "measured" or "assayed"

Line 184-185: I find this sentence awkwardly written and maybe a bit inaccurate. Maybe something like "Compared to BAHD acyltransferases, EpCAS required longer incubation times for accumulation of its chicoric acid product to become apparent, which is typical of...."

Line187: "for" rather than "against".

Line 191: I think this sentence should include why the pH optimum matters. Maybe append to the sentence "consistent with its vacuolar location."

Line 275: delete "of"

Line 316: should be "vacuoles".

Line 316: I think the first part of this sentence is a bit confusing as written. Maybe "EpCAS activity was relatively high (>80% of the maximum) over the pH range of 3.6 to 4.9, which should...."

Line 341: I would use "Additionally" instead of "Meantime"

Line 343: "better understand" instead of "to understand better"

Line 357: Please clarify. What's described sounds like a dark treatment. If so, how long was it?

Line 387: "was as follows: 2% B...", and insert space between 0.5 and min.

Line 408: check spacing

Line 418: I think you should say "Generation" rather than "Construction". If the production of hairy roots was carried out in a manner similar to the reference cited, this section could be shortened considerably.

Line 419: Please provide citations for plasmids that are being used here and elsewhere in the

manuscript.

Line 454: What do you mean by "powdered"? Do you mean the tissue was frozen in liquid nitrogen and ground with a mortar and pestle?

Line 457: There are many formulations of protease inhibitor cocktail. Please specify what was used.

Line 459: what type of membrane? Nylon?

Line 486: I'd suggest "... lane was cut out and processed for protein MS identification as detailed below."

Line 489-490: This seems like a citation to the procedure or more detail is needed.

Line 516: Perhaps "T-cep" should be spelled out.

Line 554: "donors"

Line 583: "other steps" is too vague. I'd suggest "The fractions were prepared and desalted as described for...."

Line 607-609: How much buffer was used for how much cell mass/culture? What was used for "shaking" to disrupt the cells? A bead beater?

Line 611: what type of membrane?

Line 618: How the cloning was done is not very clear to me from this. Is there a reference for the strategy used?

Line 634-635: It's not at all clear how this range of pH was achieved with the specified sodium and potassium phosphate.

Line 661: It might be helpful to the reader to include what species each of these is from.

Line 667: If this methodology is well established, a citation could be provided and the section shortened.

Line 704: Should give a citation for the vector. As a reader, I also find it helpful for authors to indicate what promoter is driving the overexpression.

Line 706-708: It is not clear how the RNAi constructs were made. Was an intermediate vector of some sort used? Is this simple antisense or hairpin RNAi? If the latter, is there an intron present to facilitate hairpin formation?

Line 710: "selected" instead of "screened"

Line 721: I don't think the delta should be superscript

Line 727: Please provide a citation for the vector

Line 729-736: Again, if this is well established methodology, a citation could be provided and the section shortened.

Line 986: "abundance" instead of "contents"

Line 991: "methods"

Line 1011: perhaps "visualized" instead of "makred"

Reviewer #2 (Remarks to the Author):

The revision is ok by now- as far as my issues are concerned

Reviewer #3 (Remarks to the Author):

In the revised version of the manuscript the authors have addressed the critical points I made in my first review.

The insertion of SDS PAGE pictures in Supplementary Figures 1 and 3 illustrates the enzyme purification from plant material and confirms the quality of heterologously produced enzymes used for in vitro activity assays.

The discussion chapter has largely been rewritten considering the points I made, pH optimum of EpCas, transport of metabolic precursors as well as alternative metabolic pathways.

English editing has increased the quality of the manuscript considerably.

Reviewer #1 (Remarks to the Author):

The authors have improved the manuscript with this revision and have addressed the main concerns I raised in my previous review. I am happy with the over all content of the manuscript, but do believe a few items still need to be addressed. I find the Methods section somewhat uneven, with sometimes too little detail given, and sometimes perhaps too much (for example where methods are relatively well established and could likely be conveyed via a citation). I'd suggest authors review that entire section to make sure appropriate level of detail is provided. I provide some examples below but it may not be exhaustive. There are also some places where the grammar and language use could be improved.

Response:

Thank you again for these valuable comments of our work. We present point-to-point responses below.

Specific comments:

Line 28: delete “be”

Response:

Corrected (line 28).

Lines 75-77: You can't really say this pathway only exists in Echinacea unless you examine every species! I'd suggest something like "... biosynthetic route has not been previously described and might exist only in Echinacea and it is likely...."

Response:

Corrected as the Reviewer 1 suggested: "This unique chicoric acid biosynthetic route has not been previously described and might exist only in Echinacea and it is likely that chicoric acid biosynthesis in other species evolved by convergent catalytic mechanisms." (line 74-75).

Line 175: I'd suggest replacing “traced” with “measured” or “assayed”

Response:

Corrected (line 174).

Line 184-185: I find this sentence awkwardly written and maybe a bit inaccurate. Maybe something like “Compared to BAHD acyltransferases, EpCAS required longer incubation times for accumulation of its chicoric acid product to become apparent, which is typical of....”

Response:

Corrected as the Reviewer 1 suggested: “Compared to BAHD acyltransferases, EpCAS required longer incubation times for accumulation of its chicoric acid product to become apparent...”(line 183-184).

Line187: “for” rather than “against”.

Response:

Corrected (line 186).

Line 191: I think this sentence should include why the pH optimum matters. Maybe append to the sentence “consistent with its vacuolar location.”

Response:

Corrected (line 191).

Line 275: delete “of”

Response:

Corrected (line 274).

Line 316: should be “vacuoles”.

Response:

Corrected (line 313).

Line 316: I think the first part of this sentence is a bit confusing as written. Maybe “EpCAS activity was relatively high (>80% of the maximum) over the pH range of 3.6 to 4.9, which should....”

Response:

Corrected as the Reviewer 1 suggested: “EpCAS activity was relatively high (>80% of the maximum) over the pH range of 3.6 to 4.9, ...” (line 313-314).

Line 341: I would use “Additionally” instead of “Meantime”

Response:

Corrected (line 339).

Line 343: “better understand” instead of “to understand better”

Response:

Corrected (line 340).

Line 357: Please clarify. What’s described sounds like a dark treatment. If so, how long was it?

Response:

As requested, we have added extra descriptions to the methodology part: “The seeds were plated on MS solid medium in the dark (covered with aluminum foil paper) for two weeks at 23 °C before germination.” (line 354-356).

Line 387: “was as follows: 2% B...”, and insert space between 0.5 and min.

Response:

Corrected (line 384-386).

Line 408: check spacing

Response:

Checked (line 406).

Line 418: I think you should say “Generation” rather than “Construction”. If the production of hairy roots was carried out in a manner similar to the reference cited, this section could be shortened considerably.

Response:

Corrected (line 426). As requested by the editor and journal policy, we have kept the detailed description in the methodology part.

Line 419: Please provide citations for plasmids that are being used here and elsewhere in the manuscript.

Response:

Added (Reference 50) (line 417).

Line 454: What do you mean by “powdered”? Do you mean the tissue was frozen in liquid nitrogen and ground with a mortar and pestle?

Response:

We have added more description to this part: “The plant material was frozen in liquid nitrogen and ground into fine powder. One gram of powdered material was extracted with 4 mL extraction buffer” (line 451-452).

Line 457: There are many formulations of protease inhibitor cocktail. Please specify what was used.

Response:

We have added product information: (product No.: 4693132001, Roche) (line 455).

Line 459: what type of membrane? Nylon?

Response:

We have added the type of membrane: polyethersulfone (PES) (line 458).

Line 486: I’d suggest “... lane was cut out and processed for protein MS identification as detailed below.”

Response:

Corrected as “When all the samples had entered the separation gel, each lane was cut out and processed for protein MS identification as detailed below.” (line 483).

Line 489-490: This seems like a citation to the procedure or more detail is needed.

Response:

We have added a citation (Reference 52) (line 489).

Line 516: Perhaps “T-cep” should be spelled out.

Response:

Corrected. We changed “T-cep” into “tris(2-carboxyethyl)phosphine” (line 513).

Line 554: “donors”

Response:

Corrected (line 551).

Line 583: “other steps” is too vague. I’d suggest “The fractions were prepared and desalted as described for....”

Response:

Corrected (line 583-584).

Line 607-609: How much buffer was used for how much cell mass/culture? What was used for “shaking” to disrupt the cells? A bead beater?

Response:

We have added more descriptions to this part: “Cells were resuspended in the same volume of Ni-NTA lysis buffer (50 mM NaH₂PO₄, 300 mM NaCl and 10 mM imidazole, pH 8.0) and broken using acid-washed glass beads (425-600 μm, Sigma) by shaking in a grinding machine (60 Hz, 60s for 5 times) and then centrifuged at 5,000 × g at 4 °C for 10 min.” (line 603-605).

Line 611: what type of membrane?

Response:

Details added: “PES” (line 608).

Line 618: How the cloning was done is not very clear to me from this. Is there a reference for the strategy used?

Response:

We have added a citation (Reference 56) (line 616).

Line 634-635: It’s not at all clear how this range of pH was achieved with the specified sodium and potassium phosphate.

Response:

We have rewritten this part into: “For determination of the effects of pH, starting solution (10 mM Na₂HPO₄ and 2 mM KH₂PO₃) was adjusted into different pH (ranging from pH 1.6 to 10.7) by adding 1M HCl or 1M NaOH”(line 630-632).

Line 661: It might be helpful to the reader to include what species each of these is from.

Response:

Added (line 658-675).

Line 667: If this methodology is well established, a citation could be provided and the section shortened.

Response:

A citation has been added (Reference 60) (line 682)

Line 704: Should give a citation for the vector. As a reader, I also find it helpful for authors to indicate what promoter is driving the overexpression.

Response:

We have added a citation for the vector (Reference 50) and the information about 35S promoter (line 708).

Line 706-708: It is not clear how the RNAi constructs were made. Was an intermediate vector of some sort used? Is this simple antisense or hairpin RNAi? If the latter, is there an intron present to facilitate hairpin formation?

Response:

We have added more detail description and a citation (Reference 68) (line 712-713).

Line 710: “selected” instead of “screened”

Response:

Corrected (line 714).

Line 721: I don't think the delta should be superscript

Response:

Corrected (line 724).

Line 727: Please provide a citation for the vector

Response:

Citation (Reference 69) added (line 731).

Line 729-736: Again, if this is well established methodology, a citation could be provided and the section shortened.

Response:

Citation (Reference 69) added (line 731).

Line 986: “abundance” instead of “contents”

Response:

Corrected (line 987).

Line 991: “methods”

Response:

Corrected (line 994).

Line 1011: perhaps “visualized” instead of “makred”

Response:

Corrected (line 1025).

Special thanks for your generous suggestions!

Reviewer #2 (Remarks to the Author):

The revision is ok by now- as far as my issues are concerned

Thank you!

Reviewer #3 (Remarks to the Author):

In the revised version of the manuscript the authors have addressed the critical points I made in my first review.

The insertion of SDS PAGE pictures in Supplementary Figures 1 and 3 illustrates the enzyme purification from plant material and confirms the quality of heterologously produced enzymes used for in vitro activity assays.

The discussion chapter has largely been rewritten considering the points I made, pH optimum of EpCas, transport of metabolic precursors as well as alternative metabolic pathways.

English editing has increased the quality of the manuscript considerably.

Thank you!

Other changes:

In Fig. 2a and supplementary Fig. 2, CcHCT, CAJ40778 has been corrected to CaHCT, CAJ40778.

In Fig. 5a, *Chysanthemun* has been corrected to *Chrysanthemum*.